# Prenylated Flavonoids in Topical Infections and Wound Healing

**DOI:** 10.3390/molecules27144491

**Published:** 2022-07-13

**Authors:** Alice Sychrová, Gabriela Škovranová, Marie Čulenová, Silvia Bittner Fialová

**Affiliations:** 1Department of Natural Drugs, Faculty of Pharmacy, Masaryk University, Palackého 1946/1, 612 00 Brno, Czech Republic; skovranovag@pharm.muni.cz (G.Š.); culenovam@pharm.muni.cz (M.Č.); 2Department of Pharmacognosy and Botany, Faculty of Pharmacy, Comenius University in Bratislava, Odbojárov 10, 832 32 Bratislava, Slovakia

**Keywords:** antibacterial, anti-inflammatory, antioxidant, cytotoxicity, mastitis, MRSA, nanotechnology, prenylated flavonoids, *S. aureus*, skin, wound healing

## Abstract

The review presents prenylated flavonoids as potential therapeutic agents for the treatment of topical skin infections and wounds, as they can restore the balance in the wound microenvironment. A thorough two-stage search of scientific papers published between 2000 and 2022 was conducted, with independent assessment of results by two reviewers. The main criteria were an MIC (minimum inhibitory concentration) of up to 32 µg/mL, a microdilution/macrodilution broth method according to CLSI (Clinical and Laboratory Standards Institute) or EUCAST (European Committee on Antimicrobial Susceptibility Testing), pathogens responsible for skin infections, and additional antioxidant, anti-inflammatory, and low cytotoxic effects. A total of 127 structurally diverse flavonoids showed promising antimicrobial activity against pathogens affecting wound healing, predominantly *Staphylococcus aureus* strains, but only artocarpin, diplacone, isobavachalcone, licochalcone A, sophoraflavanone G, and xanthohumol showed multiple activity, including antimicrobial, antioxidant, and anti-inflammatory along with low cytotoxicity important for wound healing. Although prenylated flavonoids appear to be promising in wound therapy of humans, and also animals, their activity was measured only in vitro and in vivo. Future studies are, therefore, needed to establish rational dosing according to MIC and MBC (minimum bactericidal concentration) values, test potential toxicity to human cells, measure healing kinetics, and consider formulation in smart drug release systems and/or delivery technologies to increase their bioavailability.

## 1. Introduction

The skin is the largest organ of the animal and human body and protects the internal organs from a variety of injuries as well as infectious agents. The microbiota of the skin is composed of bacteria, fungi, and viruses. Together, they form a complex ecosystem that plays a role in the defence against pathogens and in the development of the host’s immune system. Once the skin barrier is breached, the originally commensal bacteria become pathogens. They cause persistent inflammation and delay healing, leading to the development of chronic wounds typical of diabetics, immobile patients, and the elderly [1]. 

In ancient times, herbal substances were used singly or in combination with animal products, such as honey, to treat wounds. Over the centuries, therapeutic approaches were optimized until it was found that the most important thing was to prevent bacterial contamination, to maintain a moist environment in the wound, but at the same time, to absorb the exudate and exchange gases [2]. Therefore, various forms of wound dressings, such as films, hydrocolloids, hydrogels, and micro-/nanofibers, have been developed from natural and synthetic biomaterials that have the desired properties [3]. Nowadays, considerable attention is paid to the development of innovative wound dressings loaded with natural substances with therapeutic properties, such as demulsifying, emollient, re-epithelializing, astringent, antimicrobial, antioxidant, and anti-inflammatory activities, to accelerate and improve the wound healing process [2].

Data from recent years indicate that there is increasing interest in terpenes and flavonoids regarding their antimicrobial activity [4]. The aim of this review article is to provide an up-to-date overview of potent antibacterial prenylated flavonoids that contain both flavonoids and terpenes in their structure. In addition, we have highlighted those that have not only potent antibacterial effect but also anti-inflammatory, antioxidant activities along with low cytotoxicity, thus meeting the requirements to become wound healing agents.

## 2. Methods

The first step was a thorough search for scientific papers containing the terms “wound dressings”, “wound healing”, and “antibacterial” together with the keywords “flavonoids”, “polyphenols”, and “prenylated flavonoids”. The main criterion for selecting suitable compounds was the value of the MIC, which was used to quantify the effect. Natural products that showed similar activity to the reference substances or an MIC of up to 32 µg/mL were considered promising. Publications using a method other than the microdilution/macrodilution broth method described by CLSI or EUCAST and using nonstandardised CFU (colony-forming units) were excluded. The second criterion was activity against pathogens responsible for skin infections and affecting the wound microenvironment. A new search was then conducted for articles containing the terms “anti-inflammatory”, “antioxidant”, and “cytotoxic” with “the specific name of the compound”. The set of scientific papers was independently evaluated by two reviewers, and the results were processed into tables. The scientific databases Science Direct, Web of Science, and Google Scholar were used to collect scientific papers published between 2000 and 2022.

## 3. Microenvironment of Skin Wounds

In the uninjured skin, the epidermis is the outer impermeable layer that withstands the harsh external environment. Skin repair requires the intricate synchronization of several different cell types in sequential steps [5]. Immediately after injury, an inflammatory phase (1), together with aggregation and coagulation (2), restores homeostasis, stops bleeding, and prevents infection. In the first two days, mainly neutrophils are recruited to phagocytose cell debris and bacteria. They also release the growth factors. The healing process continues with the proliferation and matrix repair phase (3), which is controlled by lymphocytes, fibroblasts, macrophages, and endothelial cells. The final phase, the longest, is epithelialisation and remodelling of scar tissue (4) [5,6,7]. This healing process can be disrupted by bacteria in many ways. Contamination of the wound alters the lactate deposition, pH, and expression of proinflammatory cytokines, leading to a persistent inflammatory state with excessive levels of ROS (reactive oxygen species), toxins, and proteases [8]. The healing process is delayed as fibroblasts, growth factors, and matrix components (collagen, elastin, and fibrin) are degraded due to this adverse environment. The relationship between microbial colonisation and delayed wound healing is not yet fully understood [9], but bacterial colonisation is considered a major cause of chronic inflammation [10]. Chronic wounds, that is, those that have a biological or physiological reason for impaired healing, account for 60–80% of all infectious diseases in humans [9].

## 4. Microbiology of Skin Wounds

Microbes play an important role in influencing the wound microenvironment and, thus, wound healing. The typical sign of an infected wound is the massive proliferation of bacteria and the initiation of a host response [11]. The majority of infected wounds are contaminated with bacteria from the surrounding environment, that is, the commensal microbiota present on the skin [10]. Usually, skin infections are caused by *S. aureus*, including MRSA (methicillin-resistant *Staphylococcus aureus*). Staphylococci constitute a major group of bacteria inhabiting the skin, skin glands, and mucous membranes of humans, other mammals, and birds [12]. Other pathogens include *Streptococcus pyogenes* [13], *Pseudomonas aeruginosa*, *Escherichia coli*, *Acinetobacter* spp., and coagulase-negative staphylococci, including Staphylococcus epidermidis and *Staphylococcus lugdunensis* [9]. The first colonisers are obviously staphylococci, as their optimum growth is at a pH of around 7. Later, the wound is colonised by bacteria that can survive in a wider pH range (e.g., *P. aeruginosa* and *Enterococcus faecalis*). Peptostreptococci occur in more alkaline chronic wounds [14]. *Malassezia* spp. are the most common colonising fungal species identified on healthy skin, in contrast to *Candida* spp., which are most common in patients with immunodeficiency or diabetes or those taking antibiotics [1]. Biofilms are a serious complication of chronic wounds. This aggregated form of variable microbiota causes delayed healing. Compared with acute wounds, it is composed of anaerobic bacteria and fungi [8]. In skin wounds, the greatest biofilm formation potential has been found in *Pseudomonas*, *Staphylococcus*, *Bacillus*, and *Moraxella* spp. [9].

## 5. Bacterial Skin Infections in Livestock

Most bacterial skin infections in animals are caused by the genus *Staphylococcus* [15]. Livestock has been significantly exposed to excessive amounts of antibiotics [16] and has thus become a reservoir for bacterial resistance genes. The current threat is livestock-associated MRSA, which is transmissible to humans [17,18]. Efforts are being made worldwide to limit the use of common antibacterial agents and to offer alternatives, such as phytochemicals, for the treatment of bacterial skin diseases in livestock. As in vitro studies have shown, the plants most commonly used for healing belong to the Fabaceae and Asteraceae families [19]. One of the most common diseases treated with herbs is mastitis in dairy cows.

### Bovine Mastitis

This is a very common disease of dairy cattle caused by physical injury or by pathogenic microorganisms. It manifests itself in the form of inflammation and destruction of milk-producing tissues, resulting in reduced milk yield and poor-quality milk. The pathogens responsible for mastitis are primarily *S. aureus*, streptococci, and Gram-negative bacteria, such as *E. coli* and *Klebsiella pneumoniae* [20,21,22], so the established treatment is still based on antibiotic therapy [23]. *S. aureus* produces degradative enzymes and toxins that irreversibly damage milking tissues. However, it does not trigger an immune response in the cow as strong as other bacteria or endotoxins; it causes milder infections, leading to chronic mastitis that lasts a few months [21,24]. In addition to evolved resistance, *S. aureus* forms biofilms that protect the bacterial community from effective treatment when antibiotics cannot reach the MIC [25]. Factors such as economic losses associated with treatment, culling of animals, reduced milk production, the risk of increasing antibiotic resistance, and antimicrobial residues in milk are putting pressure on the dairy industry to focus on alternative therapies to prevent and treat bovine mastitis [21,26]. Unfortunately, the multietiological nature of the disease makes new therapeutic approaches difficult, and for example, the use of vaccines has been declared ineffective [27].

## 6. Therapeutic Strategies of Skin Infections and Wound Healing

In fact, impaired vascular function, ischaemia, superficial debris, and necrosis are the main factors causing inadequate immune response and, consequently, contaminated chronic wounds. Excessive bacterial proliferation and biofilm formation lead to a chronic and self-perpetuating inflammatory state that alters the wound microenvironment (e.g., moisture, pH, metalloproteinases, and reactive oxygen species. Therapeutic strategies then include managing as many aspects of the microenvironment as possible [8]. Nature is considered a rich source of potential therapeutics. Secondary metabolites may help to overcome pathological wound healing through pharmacological effects directed at multiple targets. Phenolics, alkaloids, essential oils (EOs), diterpenes, triterpenes, carotenoids and saponin steroids, polyunsaturated fatty acids (PUFA), glucosinolates, and polysaccharides have been reported to have anti-inflammatory, antioxidant, antibacterial, collagen-synthesis-promoting, and skin-cell-regeneration-supporting properties [28,29,30]. These phytochemicals affect one or more phases of the healing process, generally have low toxicity and good bioavailability in the skin, and are therefore widely used in wound care [28]. The advantage of treatment with natural extracts is not only the multitarget effect, but also the synergy, for example, the potentiation of the effect of the individual compounds, which can be of natural origin, but also conventional medicines. Synergistic interaction between natural products has been reported for antibacterial, antioxidant, and anti-inflammatory activities. In summary, a natural compound should ideally fulfil the four actions considered important for the treatment of skin and soft tissue infections (antimicrobial, antioxidant, anti-inflammatory, and wound healing) [31]. Widespread practice to combat infections is based on controlling the bacterial load, which is achieved by regular cleansing of the wound and the use of antiseptics, specific antibiofilm agents, and antibiotics, mostly with a local effect. Systemic antibiotics are usually not used as they are hardly available in poorly perfused tissue [32]. Nowadays, intensive research is being conducted to develop wound dressings that prevent microbes from entering wounds and have a bactericidal effect. In recent studies, plant extracts and secondary metabolites have been incorporated into various wound dressings and tested against different Gram-positive/negative bacteria. Promising active natural agents include henna (*Lawsonia inermis*), St. John’s wort (*Hypericum perforatum*), EOs, curcumin, *Aloe vera,* and thymol [10]. Some of them have even been tested in clinical trials alone or incorporated into nanoparticles. Examples include honey, various EOs, sunflower seed oil, and tea tree oil [3].

### 6.1. Wound Dressings Loaded with Natural Compounds

Several natural metabolites are candidates for promoting wound healing. An obstacle in clinical use is usually their problematic peroral or topical bioavailability.

#### 6.1.1. Essential Oils

Volatile essential oils exhibit antioxidant, antiviral, anticancer, insecticidal, anti-inflammatory, antiallergic, and antimicrobial properties [33]. These mixtures of mostly lipophilic components are considered safe and very biocompatible. Unfortunately, their therapeutic applications are limited due to low water solubility, bioavailability, and stability [34]. Recent studies guarantee efficient treatment against *S. aureus*, *S. epidermidis*, *E. coli*, *P. aeruginosa*, and *C. albicans* when essential oils are formulated in polysaccharide-based wound dressing systems [35].

#### 6.1.2. Polyphenols 

In general, polyphenols are considered promising agents due to their antibacterial, anticancer, anti-inflammatory, and antioxidant activities. The phenolic pigment curcumin exhibits numerous biological activities, such as antibacterial, antifungal, antiviral, anti-inflammatory, antioxidant, and complex wound healing properties [36]. However, the problem lies in its hydrophobicity and poor water solubility, permeability, and overall bioavailability. Therefore, it was incorporated into various formulations and tested in preclinical studies. The results showed that the glycosylation of the hydrophobic molecule, formulation in an oleic acid polymer dressing, and conjugation of curcumin with hyaluronic acid or nanofibre mats mixed with curcumin/gelatin improved solubility and availability or prolonged release. These structural modifications have been confirmed to significantly improve the regeneration process [28]. In addition, curcumin showed proven antibacterial activity against *S. aureus*, *E. coli*, *P. aeruginosa*, *B. subtilis*, and two fungi when it was incorporated into nanoparticle formulations, and against foodborne bacteria in microcapsules [37]. The catechins of green tea are also the focus of scientific interest. The 15% sinecatechin ointment Veregen^®^ has been approved by the United States Food and Drug Administration for the treatment of external genital and perianal warts [38]. As shown in a study by Chamcheu et al. [39], the effect of catechins can be enhanced when epigallocatechin gallate (EGCG) is formulated into polymeric chitosan-based nanoparticles. Topically applied nanoEGCG showed a >20-fold dose advantage over free EGCG. Green tea rich in EGCG is the type of tea that is the most extensively used in cosmetic preparations, improving skin and hair conditions [40]. Calcium, barium, and zinc alginate matrices can also form a catechin transport system that guarantees the ability to reach therapeutically relevant concentrations on the skin surface without altering release and antioxidant capacity [41].

##### Flavonoids

The flavonoid quercetin was selected for its antibacterial, anti-inflammatory, and antioxidant activity for the relief of acne. It was converted into quercetin nanofibres, which have a large porous surface area and contain many active compounds that can easily penetrate through the skin. These quercetin patches showed antibacterial activity against *Cutibacterium acnes*, safety for skin fibroblasts, and promising efficacy in clinical trials [42]. Film- and foamlike structures of N-carboxybutylchitosan (CBC) and agarose were prepared and characterised to investigate their potential application as topical membranous wound dressings. The polymeric biomaterials were loaded with quercetin and thymol, which have anti-inflammatory and anaesthetic properties, respectively, either individually or as a mixture of these two substances. Quercetin showed a more sustained release profile, which can be justified by its higher molecular volume and lower water solubility, as well as by the specific favourable interactions between quercetin and CBC [43]. The studies not only address the incorporation of quercetin into semisolid bases, such as amphiphilic creams and acidic carbomer gels, but also investigate the influence of additives (propylene glycol and polyethylene glycol) on its release and skin retention. For quercetin and chrysin, propylene glycol is a suitable absorption accelerator [44]. In a study by Roy et al. [45], the slow release of quercetin from chitosan nanoparticles prolonged the antioxidant activity of quercetin compared with its free form, whose antioxidant activities were depleted much faster. Another type of controlled delivery system, polymeric nanoparticles, increases also the antiradical and chelating properties of quercetin and catechin [46]. Interesting results were provided by a study by Hou et al. [47], in which the epidermal permeability barrier function was improved by the flavonoid apigenin. The unspecified mechanisms by which apigenin benefits the skin are stimulation of epidermal differentiation, lipid synthesis and secretion, and cutaneous antimicrobial peptide production. The flavonoids hesperidin and naringin from citrus fruits were loaded into green synthesised nanoparticles stabilised by plant gums and tested only in vitro. Nevertheless, they showed activity against MRSA and neuropathogenic *E. coli* K1 and reduced bacterial-mediated host cell cytotoxicity without toxic effect on tested human cells [48]. There are a limited number of studies with prenylated flavonoids. Four flavanones purified from the leaves of *Eysenhardtia platycarpa* Pennell and Saff. were vehicleised in nanoscale systems, particularly nanoemulsions and polymeric nanoparticles. Further in vitro release, ex vivo permeation, and in vivo anti-inflammatory studies showed a consistent release profile over time, a steady increase in flavanones in the skin permeation test, and a substantial anti-inflammatory effect [49].

### 6.2. Therapeutic Strategies for Bovine Mastitis

Phytochemicals are known to be the main source of antibiotics [50]. Many recent in vitro/in vivo studies conducted in bovine mastitis with plant-derived compounds highlight the advantages of herbal therapy. These include low toxicity, anti-infective activity against a broad spectrum of bacteria without induction of resistance even after prolonged exposure [51], and simultaneous anti-inflammatory and antioxidant activity [21]. In addition, natural anti-infectives can be used in combination with antibiotics, acting as efflux pump inhibitors, preventing biofilm formation or targeting specific bacterial virulence factors [22,52]. Numerous plants from different families have demonstrated efficacy in the healing process [19,21,22] and pure compounds also combated mastitis pathogens in livestock while attenuating inflammation. Promising compounds included the flavonoid baicalein and the monoterpene phenol thymol. A strong inhibitory effect was shown by mixtures of EOs, which probably act synergistically [21]. Their effect was confirmed in cattle when EO mixtures were applied in the form of sprays and intramammary infusions or during udder massage [19]. The positive results obtained for several plants, such as *Moringa oleifera* leaf extract, *Eucalyptus globulus* leaf extract, and *Juglans regia* plant extract, suggest a sustainable treatment alternative replacing antibiotics [21].

## 7. Flavonoids as Effective Anti-Infective and Wound Healing Agents

In the previous chapters, various flavonoids were mentioned as promising therapeutic agents suitable for wound healing. Flavonoids are widely used natural phenolic compounds. Their structure consists of a 2-phenyl-benzo-γ-pyran nucleus comprising two benzene rings. Depending on the degree of unsaturation and oxidation, they are classified into different subclasses [53]. In plants, they fulfil important functions, mainly defensive and regulatory. One of these functions is protection against bacterial and fungal pathogens. This ability has been confirmed by numerous in vitro studies. They have shown that flavonoids not only target the bacterial cell directly but also inhibit virulence factors and biofilm formation, reverse antibiotic resistance, or act synergistically with antibiotics [54]. Considering these properties, they have become patterns for semisynthetic or synthetic flavonoids combating microorganisms with MICs below 1 µg/mL. In addition to hydroxyl groups, they have been modified with halogens or other heteroatomic rings, such as pyridine, piperidine, or 1,3-dithiolium cations [55]. 

### 7.1. Flavonoids as Candidates for Therapy of Skin Lesions

Flavonoids have been shown to be excellent natural agents useful in the treatment of various skin lesions with minimal side effects [29,56,57]. Topical application is the best option for targeted use due to their lipophilic nature [57]. On the other hand, the polyhydroxyl structure determines their antibacterial, antifibrotic, antioxidant, and anti-inflammatory properties. Twenty-four structurally different flavonoids have shown the ability to accelerate healing, with quercetin, epigallocatechin gallate, and naringenin being the most studied [56]. The results showed that flavonoids decreased the levels of inflammatory mediators, such as prostaglandin E2 (PGE2), leukotriene B4 (LTB-4), interleukin 1β (IL-1β), tumour necrosis factor α (TNF-α), interleukin 6 (IL-6), interferon γ (IFN-γ); increased anti-inflammatory mediators, especially interleukin 10 (IL-10); downregulated the expression of nuclear factor kappa B (NF-κB); and inhibited the activity of cyclooxygenase (COX). Flavonoids affected cell proliferation, migration, differentiation, and angiogenesis by increasing the expression of matrix metalloproteinases 2, 8, 9, and 13. Vascular endothelial growth factor (VEGF), the main molecule regulating vascular growth, was also increased by various flavonoids. The formation of ROS significantly delays wound healing, but should be arrested by flavonoids, which increase the levels of common antioxidant enzymes. The limitation for clinical use is their low bioavailability, which is now solved by the formation of nanostructures that offer better stability, solubility, ability to cross the skin barrier, site-specific delivery, better pharmacokinetic parameters, and, in addition, a reduction in toxicity and side effects. These novel drug delivery systems in the form of nanoparticles, lipid nanocapsules, microparticles, microsponges, and so on, enable the uptake of both hydrophilic and lipophilic compounds and can be formulated into gels, creams, and other dosage forms [56,57]. Lipophilicity is a key factor of plant flavonoids against Gram-positive bacteria. The mechanism of action of flavonoids against Gram-positive bacteria likely involves the damage of phospholipid bilayers, the inhibition of the respiratory chain or adenosine triphosphate (ATP) synthesis, or some others [58].

### 7.2. Prenylated Flavonoids

They are a subclass of flavonoids modified with at least one lipophilic side chain of varying length. They attract the attention of scientists because of their promising biological activities, such as antibacterial, antifungal, estrogenic, immunosuppressive, anticancer, anti-inflammatory, antioxidant [59], antiviral, larvicidal, osteogenic, antiallergic, and cytotoxic [60]. They are found in roots, barks, seeds, and buds [61] of nontoxic or even medicinal and food plants [59]. Prenylated flavonoids have been found in the families Moraceae, Fabaceae, Cannabaceae, Guttiferae, Rutaceae, Paulowniaceae, Umbelliferae [62], Euphorbiaceae [59], Celastraceae [63], Asteraceae [64], and Thymelaeaceae [65]. Some species, such as *Artocarpus heterophyllus*, *Broussonetia papyrifera*, *Epimedium brevicornum*, *Glycine max*, *Glycyrrhiza glabra*, *Humulus lupulus*, and *Morus alba*, and propolis serve as fruits or vegetables, functional foods, or medicines in the daily diet [66]. Among the prenylated flavonoids, *C*-prenylated chalcones/dihydrochalcones, flavanones, flavones, flavonols, and isoflavones or, less frequently, *O*-prenylated forms occur. These structures are substituted with 3,3-dimethylallyl, 1,1-dimethylallyl, geranyl, lavandulyl, and farnesyl side chains, which can be modified by oxidation, reduction, dehydration, and/or cyclisation [61]. Several studies have shown that the prenyl component offers several advantages compared with parent flavonoids. In general, it causes a higher affinity to the cell membrane at the target site. Prenylated flavonoids are known to be potent P-glycoprotein inhibitors, and these abilities condition greater health-promoting properties [67]. In the case of antibacterial and enzyme inhibitory or enhancing functions, prenylation increases lipophilicity, leading to increased affinity for biological membranes and enhanced interaction with target proteins [60,68]. Cytotoxic tests showed that prenylated flavonoids have a higher binding energy in contrast to simple flavonoids [60]. Many prenylated flavonoids fulfil the assumption that they target certain diseases with an effective dose, but without having a toxic effect on their own cells. Therefore, these secondary metabolites are being intensively researched as candidates for novel dietary supplements or drugs [59]. The effect of prenylated flavonoids in the body and their pharmacokinetics after oral administration are well described. Prenylated modifications play a crucial role in absorption, tissue distribution, and metabolism. The lipophilic portion worsens the transport in the intestine into the internal circulation but, on the other hand, improves the incorporation of the prenylated flavonoids into the tissue-forming cells. Compared with the parent flavonoids, the prenylated forms are detected in tissues to a greater extent, and their accumulation lasts longer. This may indicate their difficult elimination from tissue-forming cells via efflux pumps or a low rate of glucuronidated forms. In summary, prenylation enhances various biological effects but also carries the risk of potential side effects, especially with long-term dietary intake [67]. There is less information on the topical use of prenylated flavonoids. Dong et al. [69] tested in vivo the anti-inflammatory activity of sophoraflavanone G, the prenylated flavanone presented in *Sophora flavescens*, with effects observed after oral and topical administration. Although the potencies of inhibition were far below those of the reference drug prednisolone, sophoraflavanone G showed higher anti-inflammatory activity when applied topically [69].

### 7.3. Mechanisms of Antibacterial Activity

In general, the structure of 2-phenyl-1,4-benzopyrone is crucial for the antibacterial activity of flavonoids and prenylated flavonoids. The available reports on the mechanisms of antibacterial action have led to different results. It seems possible that flavonoids may not only affect one specific target but also influence several cellular processes. Existing research has suggested that antibacterial activity may be caused by the following mechanisms [70].

#### 7.3.1. Direct Interaction with Bacterial Cell

For apigenin and quercetin, the inhibition of cell wall synthesis has been observed via reversible inhibition of D-alanine–D-alanine (D-Ala–D-Ala) ligase, the essential enzyme important for the ligation of D-Ala–D-Ala in the completion of peptidoglycan precursors [71]. Alteration of cell membrane permeability and damage to membrane functions were found for several flavonoids. Direct damage to the bacterial cytoplasmic membrane using hydrogen peroxide were the first mechanisms of action attributed to various flavan-3-ols [72], flavolans [73,74,75], and green tea catechins [75,76]. Flavonoids generate hydrogen peroxide by releasing a hydrogen from their pyrogallol or catechol structure to oxygen via a superoxide anion radical [77]. On the other hand, catechins can cause membrane fusion, leading to leakage of intramembranous material and aggregation. Inhibition of membrane function has also been discovered for galangin [78] and quercetin [79]. Sophoraflavanone G showed that lipophilic flavonoids may be able to reduce the fluidity of the outer and inner layers of cell membranes [80]. Other mechanisms of antibacterial action have been described for retrochalcones isolated from *Glycyrrhiza inflata* as an effect on the biosynthesis of macromolecules. Retrochalcones inhibited the incorporation of thymidine, uracil, and leucine into macromolecules, such as deoxyribonucleic acid (DNA), ribonucleic acid (RNA), and proteins. They also inhibited the oxidation of nicotinamide adenine dinucleotide (NADH) and oxygen consumption in the bacterial membrane. The results suggest an inhibition of respiration between coenzyme Q and cytochrome c in the bacterial electron transport chain [81]. The mechanism of ATP synthase or hydrolysis blockade has been described. This enzyme is responsible for ATP generation through phosphorylation and photophosphorylation; therefore, the antibacterial effect is attributed to the inhibition of energy metabolism. This effect is caused by the interaction between the flavonoid and polyphenol binding pocket residues of the enzyme. The size, shape, geometry, and presence of functional groups of the compounds are crucial for the binding and inhibition of the enzyme [82]. This mechanism of action could also disrupt bacterial motility [79]. For example, seventeen flavonoids have been shown to block ATP synthase and subsequently inhibit energy metabolism in *E. coli* [82]. Another suspected mechanism has been suggested in nucleic acid synthesis. Flavonoids have been identified as promising topoisomerase I inhibitors due to their redox, structural, and steric properties. They must undergo oxidation to quinones and could then interact with the DNA topoisomerase complex [83]. The inhibition of DNA gyrase has been found for quercetin and apigenin, for example [84], and another flavonoid, rutin, could interact with topoisomerase, in particular, IV [85]. Flavones and flavonols were identified as inhibitors of helicases and thus interfere with the process of separation of two cross-linked nucleic acid strands [86]. Extensively studied catechins from green tea showed activity against *Proteus vulgaris* and *S. aureus*. The mechanism of their action was elucidated using radioactive precursors as flavonoid–DNA intercalation when DNA and protein synthesis RNA inhibition was shown [87]. In another study, EGCG was able to affect an important bacterial enzyme, dihydrofolate reductase. In addition, EGCG enhanced the effect of standard inhibitors of folic acid metabolism, such as sulfamethoxazole and ethambutol [88]. Many flavonoids can also inhibit bacterial metal enzymes due to their chelating ability [89]. Another target of action is to influence fatty acid biosynthesis. It has been published that some flavonoids (e.g., EGCG) are able to inhibit three successive enzymes: β-ketoacyl-ACP reductase (FabG), β-hydroxyacyl-ACP dehydratase (FabZ), and enoyl-ACP reductase (FabI) [90]. 

#### 7.3.2. Indirect Antimicrobial Activity

The inhibition of bacterial pathogenicity is considered one of the nonspecific mechanisms of antibacterial action. Flavonoids can inhibit the quorum sensing system, which is important for bacterial communication and regulation of virulence factors, including biofilm formation [91]. In a study by Vikram et al. [92], the citrus flavonoids apigenin, kaempferol, quercetin, and naringenin were highlighted as significant antagonists of cell–cell signalling. The apple flavonoid phloretin reduced the expression of genes involved in toxin production and fimbriae formation [93]. Other flavonoids, such as myricetin, quercetin, kaempferol, pinocembrin, catechins, and proanthocyanidins, also neutralise bacterial toxic virulence factors (e.g., hyaluronidase and α-hemolysin [54]. The unexpected discovery was the nonspecific aggregating effect of flavonoids on whole cells of bacteria. It has been postulated that the antibacterial effect of flavonoids does not target specific enzymes and may not affect enzymes at all [70,91]. Nevertheless, this bacterial cell aggregation affects membrane integrity and causes biofilm disruption [94]. In summary, studies have led to the realisation that a compound can have multiple mechanisms (see Figure 1).

### 7.4. Structure–Activity Relationship

The presence of hydrophobic and hydrophilic moieties is crucial for the antibacterial activity of flavonoids [95]. The prenyl group on the condensed pyran ring system normally controls this activity. In contrast, the presence of the prenyl group was not required for aryl substitutions [96]. The hydrophobic substituents such as alkylamino and/or alkyl chains and nitrogen- or oxygen-containing heterocyclic moieties usually enhance the antibacterial activity for all flavonoids [97]. On the other hand, modifications of the lipophilic side chain with carbonyl, hydroxyl, and methoxyl moieties and/or cyclisation of the prenyl and/or geranyl substituent reduce the activity [97,98]. The most structurally active compounds include chalcones, flavanones, and flavan-3-ols [99]. Antibacterial properties depend on several structural features typical for each class of flavonoids. In general, the hydroxyl groups are able to trigger and enhance the anti-MRSA activity of flavonoids, while the presence of methoxy units drastically decreases the antibacterial activity. Hydroxyl groups in positions 2′ of the chalcones and 5 of the flavanones (or flavones) increase activity against MRSA, while the methoxy groups have the opposite effect. Very promising anti-MRSA activity was measured for 2′(OH)-chalcone, 2′,4′(OH)2-chalcone, and 2′,4(OH)2-chalcone [100]. Considering their antibiofilm activity against MRSA, hydroxylations in positions 2′ or 4′ in the A ring and 4 in the B ring also seem to be relevant structural features. Some heterocyclic chalcone analogues have been synthesised with the result that replacing the aromatic ring B with a heterocyclic ring containing nitrogen, oxygen, or sulphur atoms does not significantly increase antibacterial activity against MSSA and MRSA [101]. On the other hand, a lipophilic substitution of ring A increases the activity [97]. Xie et al. [97] summarised that 5,7,4′-hydroxyl substitutions indicate the antibacterial activity of flavones, and their methylation decreases the activity, while flavonols seem to be better antibacterial agents than flavones. Substitutions in the A ring at positions 7 (-*O*-acyl or -*O*-alkylamino) and 5-hydroxyl have been reported to be crucial for the antibacterial activity of flavones. Favourable interactions are found in the B ring, where positions 3′ or 4′ are hydroxylated or *O*-acylated. Hydroxyl or methoxy substituents at position 6′ cause moderate antibiofilm activity [102]. Flavanones containing a saturated C3–C4 bond are considered the more promising compounds than flavones [97]. Results by Oh et al. (2011) showed that lavandulyl or isoprenyl groups at C-8 contribute to the antibacterial activity of prenylflavanone derivatives [103]. The number and position of prenyl/geranyl and hydroxyl groups determine the anti-MRSA activity of flavanones and flavanonols [99], with at least C-5, C-7, and C-4′ hydroxylations being basic requirements [98,104]. Structures with a 2′,4′- or 2′,6′-dihydroxylation of the B ring and substituted with a long-chain aliphatic group such as lavandulyl or geranyl at the 6- or 8-position showed strong activity [97]. In contrast, Tsuchiya et al. [104] investigated that the lavandulyl group was a more efficient moiety for enhancing antibacterial activity than geranyl. Dihydroflavonols generally show better activity than flavonols, and the compounds with double prenyl substituents are more active than the corresponding monosubstituted ones. Among the flavanols, especially catechins and theaflavins show antibacterial properties against common pathogenic bacteria, such as *S. aureus,* MRSA, *E. coli*, and *H. pylori* [97]. In general, oligomeric flavanols show higher activity than monomeric ones, and this rule also applies to flavan trimers compared with dimers [105,106]. The structure–antimicrobial activity relationship of flavonoids is explained in Figure 2.

## 8. Discussion

As shown in Table 1, prenylated flavonoids occur in various families and plant parts. We have mentioned promising antibacterial agents from the families Asteraceae, Celastraceae, Euphorbiaceae, Fabaceae, Moraceae, and Paulowniaceae, with Fabaceae and Moraceae being the most abundant. A separate section was devoted to propolis and substances with an unspecified source. Each table contains the plant source, the name of the compound, the type of bacterium against which the substance is active, and the MIC value. For comparison, we list the positive controls used in the tests. These tables provide a quick overview of the antibacterial activity of prenylated flavonoids, present compounds that meet the criterion of an MIC of up to 32 µg/mL, and show microorganisms responsible for impaired wound healing that are antagonised by prenylated flavonoids. The structures of the prenylated flavonoids are plotted in Figure 3, Figure 4, Figure 5, Figure 6, Figure 7, Figure 8, Figure 9, Figure 10, Figure 11 and Figure 12. The red highlighted parts of molecules indicate the difference between similar structures and might explain the structure–antibacterial activity relationship.

### 8.1. Prenylated Flavonoids with Potent Antibacterial Activity

From the present study, it is possible to highlight compounds that surpass others by their activity. These flavonoids are structurally different, and therefore, it is not the aim of this review to compare these flavonoids with each other. It is clear that *S. aureus* strains, including MSSA and MRSA, are the most sensitive to the action of prenylated flavonoids. According to Farhadi et al. [99], the most active compounds are chalcones, flavanones, and flavan-3-ols. This study confirmed that many chalcones showed promising anti-staphylococcal activity [99]. Isobavachalcone reduced the growth of MRSA strains with MICs in the range of 4–8 µg/mL, compared with control antibiotics that achieved MICs above 128 μg/mL [131]. Its activity was later confirmed by de Assis et al. [145]. Song et al. [142] tested a large number of compounds, and among them the, chalcones isobavachalcone, 4-hydroxyderricin, kanzonol C, xanthohumol, licochalcone A, licochalcone C, and licochalcone E antagonised sensitive and resistant strains with MICs between 2 and 4 µg/mL. Similar results were obtained in a study by Wu et al. [144], where licochalcone A, licochalcone C, and licochalcone E showed activity with MICs of 0.5–16 µg/mL. Licochalcones have also been successfully tested in other studies [144]. An MIC of 4 µg/mL was achieved for licochalcone A in Liu et al. [143]. The activity of licochalcone E was robust (MICs = 1–4 µg/mL) compared with the MICs of oxacillin (0.1–256 µg/mL) in Zhou et al. [146]. Several flavanones achieved very low MICs. These, isolated from *Dalea scandens* [110] and malheurans A–C, had the highest MIC of 4.6 µg/mL [112]. Kurarinone effectively inhibited MRSA and furthermore VRE with an MIC of 2 µg/mL, in contrast to ampicillin, which combated them with an MIC of 250 µg/mL [124]. The structurally similar sophoraflavanone G was successful in eradicating *S. aureus* strains with MICs in the range of 0.5–8 µg/mL compared with standard antibiotics whose MICs were several times higher (0.1–1024 µg/mL) [123,125]. Song et al. [142] determined promising activity with MICs of 1–2 µg/mL against MSSA and MRSA for euchrestaflavanone A, sophoraflavanone C, and glabrol, whose activity was also demonstrated by Wu et al. [144] against more than 20 MRSA strains with MICs of 1–4 µg/mL. Highly active geranylated flavanones include sepicanin A [129], mimulone, and variously substituted diplacones isolated from the fruit of *P. tomentosa* [98,140]. Another rich source of geranylated compounds is the root bark of *M. alba*. The MICs detected for kuwanons E and U ranged from 2 to 4 µg/mL. This plant also contains the flavones kuwanons B, C, and T and morusin, which have activity with MICs ranging from 1 to 4 µg/mL, depending on the strain used. Importantly, the results obtained are coherent and have been demonstrated by different authors [136,137,138,139]. Flavone artocarpin is characterised by antistaphylococcal activity in the range of 1–2 µg/mL, but also other flavones, such as 5′-geranyl-5,7,2′,4′-tetrahydroxyflavone, 3′-geranyl-3-prenyl-5,7,2′,4′-tetrahydroxyflavone, isoneobavaisoflavone, and 6,8-diprenylgenistein, are among the significantly active compounds with MICs of 2 to 4 µg/mL [142]. Alpinumisoflavone tested in the study by Chukwujekwu et al. [115] had an MIC of 3.9 µg/mL, while Sadgrove et al. [116] found it not as active with an MIC of 31 µg/mL. This discrepancy may be caused by the use of different *S. aureus* strains.

The selected compounds showed better antibacterial activity against a broad spectrum of microorganisms than the others. The pterocarpan erybraedin A was evaluated against skin pathogens, such as *B. cereus*, *S. aureus*, *S. epidermidis*, and *E. coli*, and showed MICs in the range of 1–2 µg/mL. The flavan eryzerin C achieved slightly lower activity with MICs of 2 to 10 µg/mL against the same microorganisms [116]. The chalcone xanthohumol showed activity against *S. epidermidis*, *S. capitis* ssp. *ureolyticus*, *S. aureus*, and MRSA with MICs of 2–4 µg/mL [148]. Angusticornin B and bartericin A, two diprenylated chalcones, were able to eliminate a broad spectrum of skin pathogens, including four *Bacillus strains*, *S. aureus*, and *E. faecalis*; two Gram-negative bacteria, *E. coli* and *P. aeruginosa*; and three *Candida* strains with very low MICs (<0.3–9.8 µg/mL) [130]. A similar spectrum of microorganisms was used in a study by Mbaveng et al. [132], where isobavachalcone strongly inhibited the growth of pathogens with MICs of 0.3–1.2 µg/mL and 4-hydroxylonchocarpin with MICs of 1.2–4.9 µg/mL despite cyclisation of the prenyl chain. Very low MICs (0.8–2 µg/mL) were obtained when testing the diprenylated flavanone lonchocarpol A against MRSA, VRE. *Faecium*, and *B. megaterium* [117]. The geranylated flavanones isolated from *P. tomentosa* not only showed activity against *S. aureus*, but also combated *Bacillus* strains and *E. faecalis* [140]. Propolin C, actually diplacone, demonstrated activity in a study by Chen et al. [141]. Kuwanons and morusin inhibited the growth of *E. faecalis*, VRE, and *B. subtilis* [137,138]. In a study by Polbuppha et al. [134], the isoflavanone lupalbigenin showed promising activity with MICs of 1–4 µg/mL against *E. faecalis*, *S. aureus*, MRSA, and *C. albicans*. The results obtained for *S. aureus* strains are similar to the MICs reported in a study by Song et al. [142]. Two flavones, artocarpin and cudraflavone C, showed activity against *C. acnes*, *S. aureus*, and *S. epidermidis* in the range of 2–4 µg/mL [128]. The rare compound containing three isoprenyl units on a modified A ring, millexatin A, together with the other isoflavones, millexatin F, auriculatin, and scandenone, showed remarkable activity against *S. aureus*, *S. epidermidis*, and *B. subtilis*. These compounds were active in the range of 2–4 µg/mL [118]. The results are consistent with a study by Polbuppha et al. [134], in which millexatin F, auriculatin, and scandenone controlled *E. faecalis*, *S. aureus*, MRSA, and *C. albicans* with MICs in the range of 2–4 µg/mL for bacteria and 2–8 µg/mL for yeast. Other results were presented in a study by Özçelik et al. [135], where scandenone was more effective against *S. aureus* and *E. faecalis* (MICs = 0.5 µg/mL). Table 2 lists the compounds that have several beneficial activities in wound healing and show anti-inflammatory and antioxidant activities in addition to antibacterial activity. Some of these compounds have reduced the pathogenicity of microorganisms through various mechanisms. For completeness and future research, data on cytotoxic activity have also been included. It is not the purpose of this review to detail the mechanisms of anti-inflammatory and antioxidant activities of prenylated flavonoids, as this information can be found in many studies (see Table 2). Let us take a closer look at the compounds with the most promising wound healing properties.

### 8.2. Multiple Active Prenylated Flavonoids as Wound Healing Agents

Artocarpin is a flavone from *Artocarpus* species (Moraceae) substituted at position 3 by prenyl and at position 6 by a (1E)-3-methylbut-1-enyl moiety [257]. Studies have shown its remarkable antibacterial activity [128,142] and its ability to act synergistically with norfloxacin, tetracycline, and ampicillin in the control of MRSA, *P. aeruginosa*, and *E. coli* [155]. Lee et al. [158] suggested its topical dose of 0.05% as a photoprotective agent in medicine and/or cosmetics due to its ability to prevent skin damage caused by UVB irradiation. Artocarpin was found to reduce scaling, epidermal thickening, and sunburn cell formation by lowering the levels of ROS and lipid peroxidation and by downregulating proinflammatory cytokines and proteins. It also inhibits tyrosinase and melanogenesis as two targets in the skin whitening process [257]. It has been evaluated in wound healing studies conducted in vitro and in vivo. Artocarpin accelerates the inflammatory phase and increases myofibroblast differentiation, fibroblast and keratinocyte proliferation and migration, collagen synthesis and maturation, re-epithelialisation, and angiogenesis [159]. However, as it shows several beneficial effects, it remains to be clarified whether artocarpin has the potential to be a therapeutic agent for the treatment of skin wounds. Its cytotoxic effect, demonstrated by different authors [156,157,160,258] in different cell lines, needs to be considered. The most important information is the cytotoxic effect on keratinocytes. Artocarpin has exerted cytotoxicity on HaCaT keratinocytes at 10 µM in in vitro studies and at 0.1% in in vivo studies, but a dose of 0.05% artocarpin has been shown to be a safe dose for topical formulation [158]. In a study by Yeh et al. [159], 1–2 µM artocarpin used in in vitro studies and 0.08% artocarpin used in in vivo studies showed no cytotoxic effects on human keratinocytes.

Diplacone, structurally 6-geranyl-3′,4′,5,7-tetrahydroxyflavanone, is also known as propolin C or nymphaeol A. Its structure has been identified as an eriodictyol with a C-10 side chain (geranyl) substituted in the C-6 position. Its promising antibacterial activity has been confirmed by several authors [98,140,141]. Moreover, diplacone inhibits *S. aureus* biofilm formation in a dose-dependent manner [107]. Diplacone exhibits anti-inflammatory activity both in vitro and in vivo through different mechanisms of action. It downregulates the expression of TNF-α and MCP-1 and upregulates the expression of zinc finger protein 36, promoting the degradation of cytokines. Its effects are even better than those of indomethacin, which has been used as a positive control [165]. It inhibits the production of NO in LPS-stimulated macrophages [173], and two independent studies showed that diplacone affects the expression of COX-2 [167] and its activity. Moreover, it was found to be a potent 5-LOX inhibitor [168]. In vivo, diplacone alleviates the symptoms of colitis and delays its onset [170]. Due to its ortho-dihydroxy functionality in the B ring, it possesses very good radical scavenging activity (e.g., as a DPPH scavenger). The geranyl side chain has no significant effect on the antioxidant activity [164,166,171]. It also shows a protective effect on H_2_O_2_-induced injury of HUVECs [259]. Due to its multiple activities, diplacone seems to be a suitable candidate for wound healing; however, its potential cytotoxic effect on normal cell lines remains to be verified, as it shows activity on cancer cells at low concentrations [169,172]. 

Isobavachalcone is a 3′-prenylchalcone purified from the families Fabaceae, Guttiferae (syn. Clusiaceae), Moraceae, Schisandraceae, and Umbelliferae (syn. Apiaceae) [260]. It shows activity against Gram-positive bacteria, mainly MSSA and MRSA [131,142,145]. Furthermore, it is able to inhibit more than 75% of MSSA and MRSA biofilm formation as effectively as vancomycin [145]. Isobavachalcone suppresses the production of nitric oxide, one of the major mediators of inflammation [153,181,183], negatively regulates inflammation-related enzymes such as iNOS [178] and 15-LOX [183], and attenuates the inflammatory response in Sephadex-induced lung injury in vivo [184]. It exhibits direct radical scavenging activity in rat liver microsomes and mitochondria [182], peroxyl radical scavenging capacities [179], and upregulation of antioxidant enzymes [184]. Isobavachalcone is selectively cytotoxic to cancer cells [180] and shows less cytotoxicity to normal cell lines, such as hepatocytes [177], foetal hepatocytes, umbilical vein endothelial cells [185], and cerebellar granule cells [186]. Finally, it is an easily synthesised substance [145].

Licochalcone A is 5-(2-methylbut-3-en-2-yl)chalcone, apparently isolated from the roots of licorice. It very effectively controls susceptible and resistant staphylococcal strains [142,143,144] and also suppresses the secretion of their enterotoxins A and B [209]. Licorice root has traditionally been used to treat inflammatory diseases. Modern studies have confirmed this effect and identified possible mechanisms of action, with licochalcone A targeting multiple levels of the inflammatory response. It inhibits the activation of transcription factors, such as NF-κB [207,213,214,217] and AP-1 [216]; suppresses TNF-α, IL-6, IL-1β cytokines [214] and the production of PGE2 and NO, and reduces the expression of iNOS and COX-2 [215]. It proves to be an effective inhibitor of NLRP3 inflammasome activity triggered by *C. acnes*, preventing the development of inflammation and exacerbation of acne lesions [210]. A moisturising cream with licochalcone A, 1,2-decanediol, L-carnitine, and salicylic acid, which was tested in a clinical trial, reduces acne lesions and prevents the development of new lesions during the maintenance phase [261]. Moreover, several studies have revealed licochalcone A as a promising anti-irritant in the management of sensitive skin [262,263,264,265]. Experimental data show that licochalcone A suppresses cell oxidation, and 2–8 μg/mL induces the expression of SOD, CAT, and GPx1 proteins [208]. It shows low cytotoxicity without haemolytic activity based on safety assessment [144]. However, this finding is not consistent with a study by Chen et al. [208], who found slightly different IC_50_ values for the same HepG2 cell line, and as usual, lower cytotoxic concentrations were observed in cancer cell lines [211]. Licochalcone A is at the forefront of scientific interest as a lead agent against various diseases, but unfavourable biopharmaceutical properties limit its therapeutic use. Fortunately, its poor solubility and potential haemolytic and cytotoxic effects were overcome in a study by Silva et al. [266], in which licochalcone A was incorporated into solid lipid nanoparticles. The penetration of licochalcone A into the epidermis can be increased with micellar vehicles coloaded with glycyrrhizic acid [267]. 

Sophoraflavanone G is a flavonoid substituted with a C-8 lavandulyl group. It eradicates resistant *S. aureus* strains with MICs disproportionately lower than the MICs for conventional antibiotics [123,125]. It enhances the effect of antibiotics when it shows additivity with ciprofloxacin, erythromycin, gentamicin, fusidic acid, and oxacillin [123] and synergy with ampicillin and oxacillin [125]. Sophoraflavanone G has been found to be a potent inhibitor of eicosanoid-forming enzymes [69,191,240]. It disrupts signalling pathways associated with inflammation, including NF-κB and MAPK [237]. In vivo, it shows higher activity when applied topically than when taken orally [69]. Unfortunately, sophoraflavanone G is considered a potent antitumour agent that inhibits cell proliferation in vitro [238,239,241] and in vivo [238].

Xanthohumol is the most abundant prenylated chalcone in hops (*Humulus lupulus* L.) and shows remarkable antistaphylococcal activity [142,147,148], which is directed against both planktonic and biofilm forms of bacteria. Sub-MIC concentrations prevent staphylococcal adhesion to abiotic surfaces, resulting in the inhibition of biofilm formation. A concentration-dependent reduction in viability up to complete eradication of the biofilm has been also observed [147]. The antibiofilm properties have been confirmed by Bogdanova et al. [148], where xanthohumol reduces the number of bacterial cells released from the biofilm, penetrates the biofilm, antagonises bacteria, and, at higher concentrations, reduces the number of surviving bacterial cells to zero. In addition, xanthohumol enhances the effect of oxacillin [147]. The multiple targets and mechanisms explain the broad anti-inflammatory effect of xanthohumol. It inhibits the production of NO by suppressing inducible NO synthase [247], decreases NF-κB activation in vitro [244,249] and in vivo [244], and inhibits the production of two proinflammatory cytokines, MCP-1 and TNF-α [252]. Finally, xanthohumol has been found to be effective in attenuating skin inflammation by inhibiting IL-12 production [254]. Xanthohumol ingestion reduced inflammation, oxidative stress, and angiogenesis and improved the wound healing process without toxicity in the tested Wistar rats [255]. In addition, it showed the ability to regulate the activities of elastases/MMPs and stimulate the biosynthesis of fibrillar collagens, elastin, and fibrillins, preventing skin ageing [268]. The ability to scavenge reactive oxygen species and influence the endogenous antioxidant system has been demonstrated in several studies in vitro [244,245,246,250] and in vivo [244,255]. Xanthohumol is considered an effective chemopreventive and therapeutic agent in cancer treatment due to its ability to inhibit carcinogenesis and metastasis. Despite these properties, it shows very little or no toxicity in normal cells, including human lung fibroblasts, primary human hepatocytes, oligodendroglia-derived cells, and human skin fibroblasts. Similar results have been obtained in in vivo assays [269]. The process of its chemical synthesis is demanding, and the overall yield is relatively low. Therefore, female inflorescences are still the main source of xanthohumol [270]. 

## 9. Conclusions

Microbially infiltrated and damaged wounds of both humans and animals could be treated with prenylated flavonoids, such as artocarpin, diplacone, isobavachalcone, licochalcone A, sophoraflavanone G, and xanthohumol, which have shown promising multiple activities required in the process of wound healing. The evidence and preliminary results suggest that further studies are warranted. In these future studies, several factors need to be considered, including rational dosing according to MIC and MBC values, potential toxicity to human cells, healing kinetics, type of wound, chronicity, timing of application of the therapeutic agent, patient condition, origin, age, combinatory effects, contact time, bacterial strain present, and so on. An obstacle in clinical use could be difficulties in administration. Such preparations need to be formulated with smart drug release systems and/or delivery technologies to be acceptable for the treatment of patients. As regards manufacturing, their hydrophobicity allows them to be efficiently combined with polymeric matrices, which are often used as wound dressings. Nanotechnology can enhance the release of these antimicrobial, anti-inflammatory, and regenerative substances, thus accelerating the body’s own healing process. In addition, prenylated flavonoids can overcome the disadvantages of current antibiotics and antiseptics (especially cytotoxicity, antibiotic resistance, and allergies) or enhance the effect of conventional drugs. From an economic point of view, thought must be given to the way in which these substances are obtained, whether by isolation from plant material or by synthetic preparation. Very often, it is difficult to obtain the pure compound in sufficient quantity, and its use is economically disadvantageous. Then enriched standardised extracts could replace the pure compounds. For example, Glabridin-40, a glabridin-enriched extract of *Glycyrrhiza glabra* (root), is widely used in cosmetic formulations as an anti-inflammatory, antioxidant, and skin whitening agent.

## Figures and Tables

**Figure 1 molecules-27-04491-f001:**
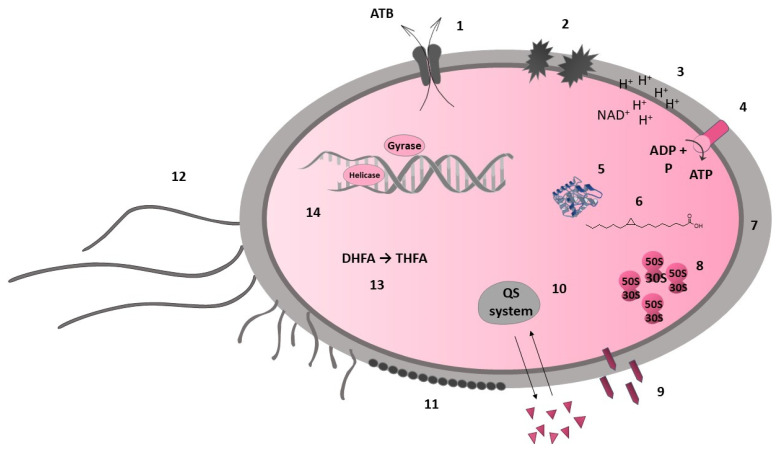
Bacterial targets of flavonoids. Cell membrane: efflux pump (1), membrane disruption (2), electron transport chain (3), ATP synthesis (4); bacterial metalloenzymes (5); fatty acid synthesis (FabG, FabZ, FabI) (6); cell wall synthesis (7): peptidoglycan, D-alanine–D-alanine ligase; protein synthesis (8): (cell envelope); nonspecific mechanism: bacterial toxic virulence factors (9), quorum sensing system (10), biofilm formation (disruption) (11), motility (12); folic acid metabolism: dihydrofolate reductase (13); nucleic acid synthesis (14): DNA gyrase, topoisomerases I and IV, helicase, DNA intercalation.

**Figure 2 molecules-27-04491-f002:**
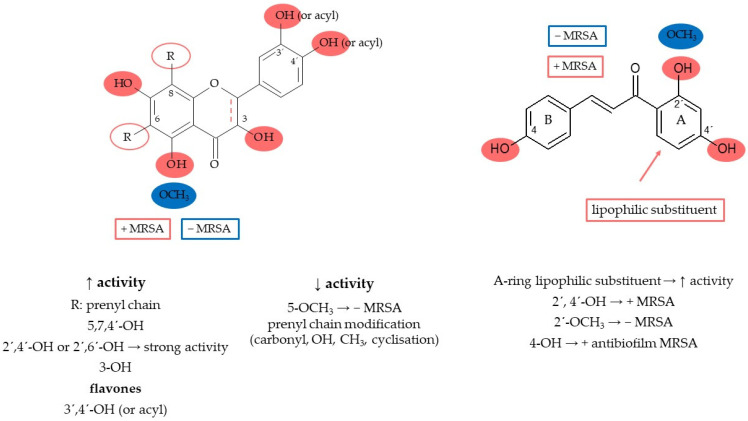
Flavonoids’ structure–activity relationship.

**Figure 3 molecules-27-04491-f003:**
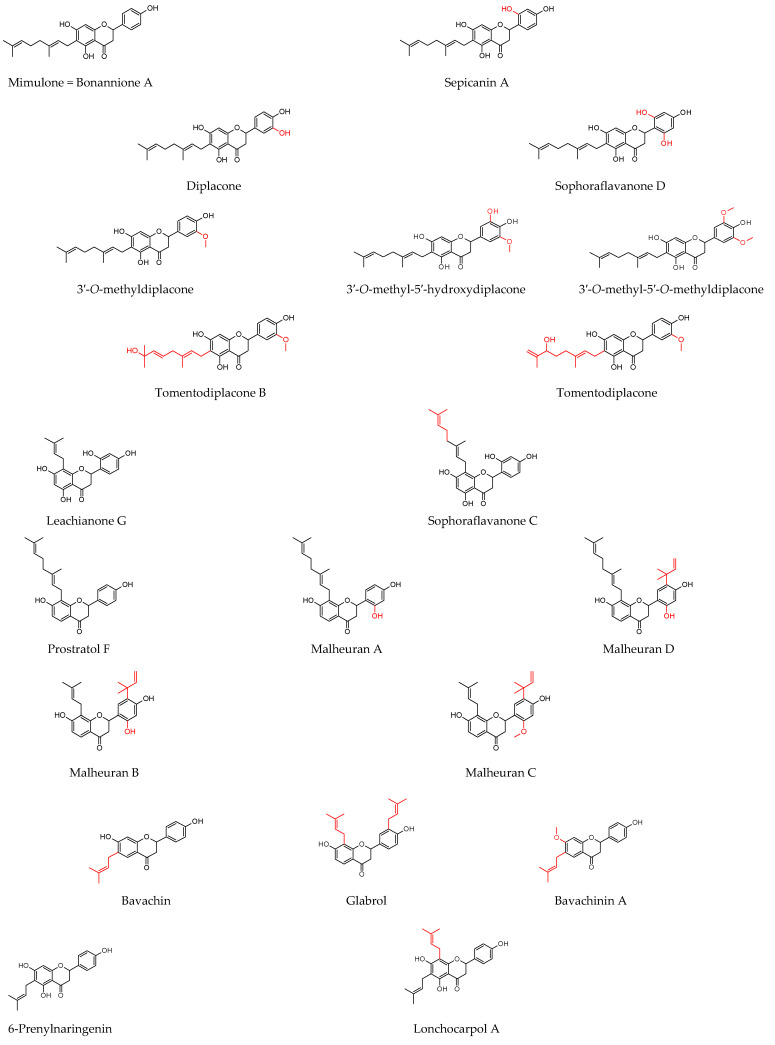
Structures of discussed flavanones.

**Figure 4 molecules-27-04491-f004:**
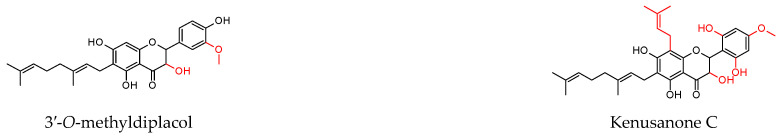
Structures of discussed flavanonols.

**Figure 5 molecules-27-04491-f005:**
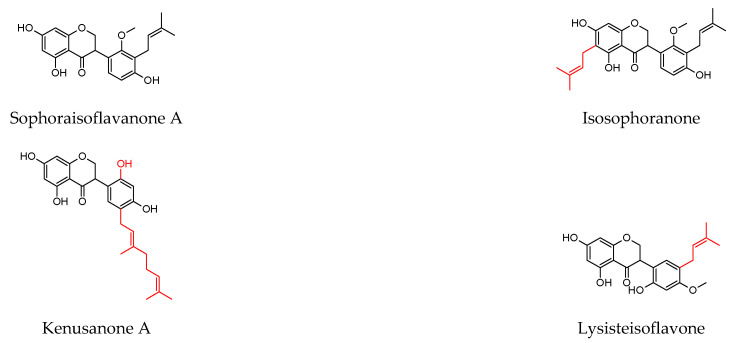
Structures of discussed isoflavanones.

**Figure 6 molecules-27-04491-f006:**
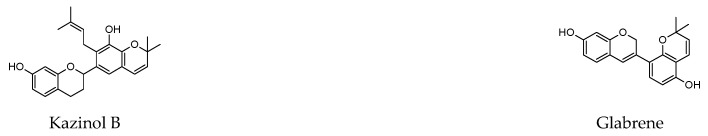
Structures of discussed flavan, isoflaven.

**Figure 7 molecules-27-04491-f007:**
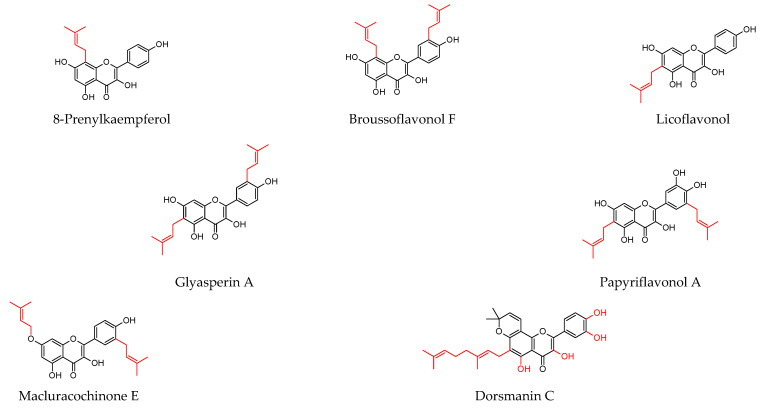
Structures of discussed flavonols.

**Figure 8 molecules-27-04491-f008:**
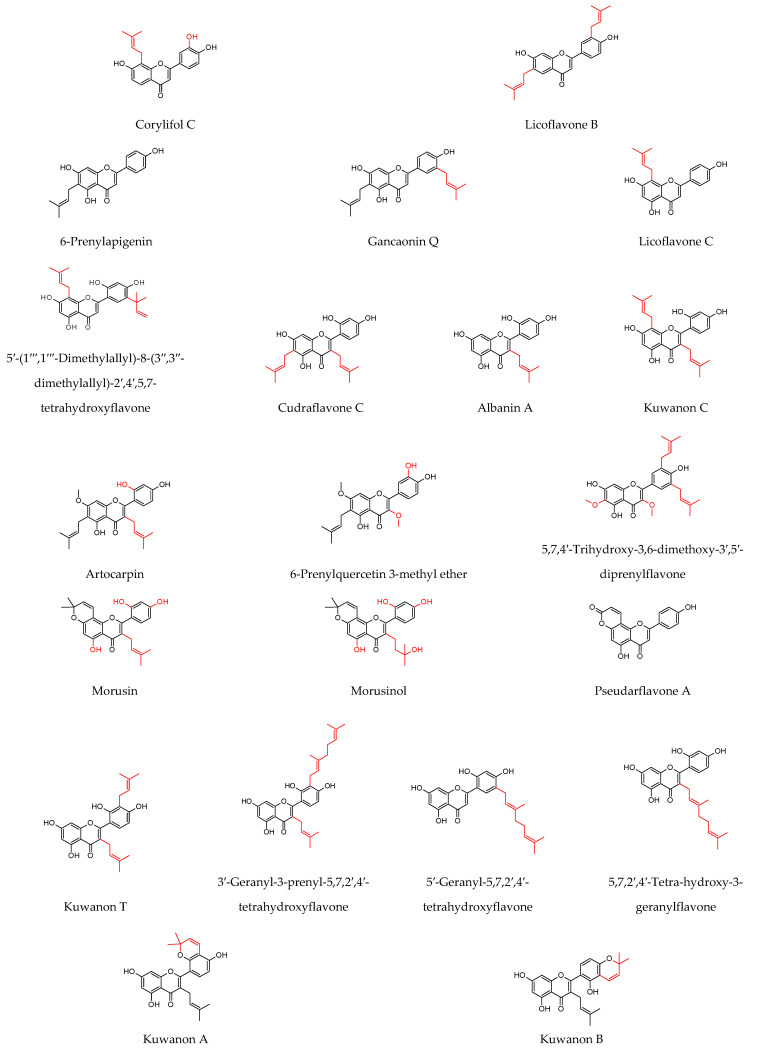
Structures of discussed flavones.

**Figure 9 molecules-27-04491-f009:**
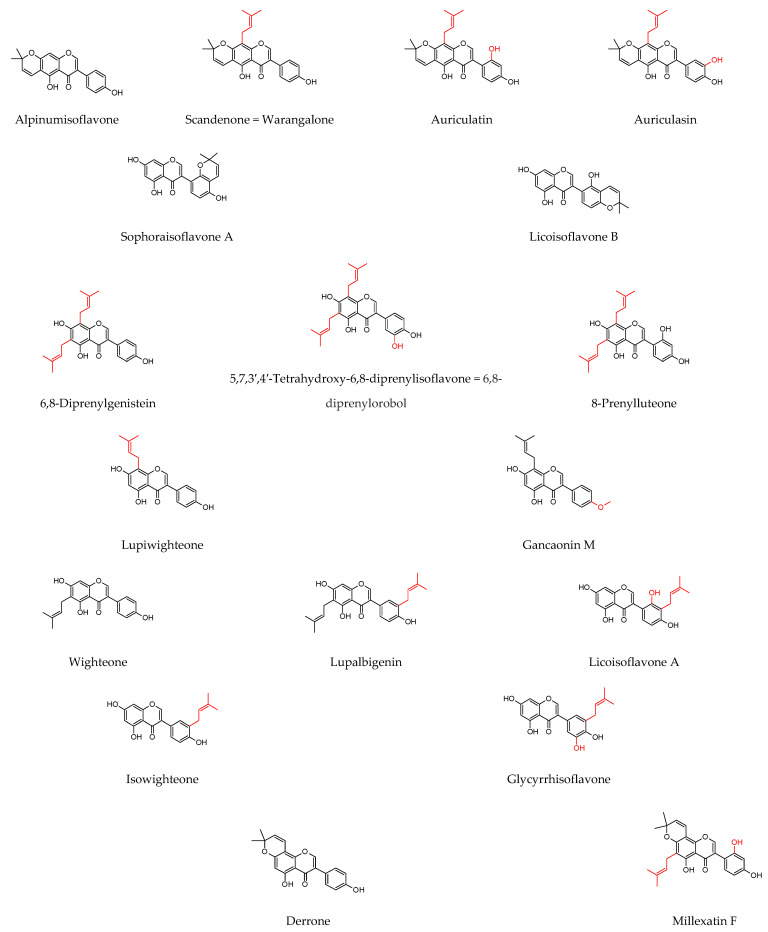
Structures of discussed isoflavones.

**Figure 10 molecules-27-04491-f010:**
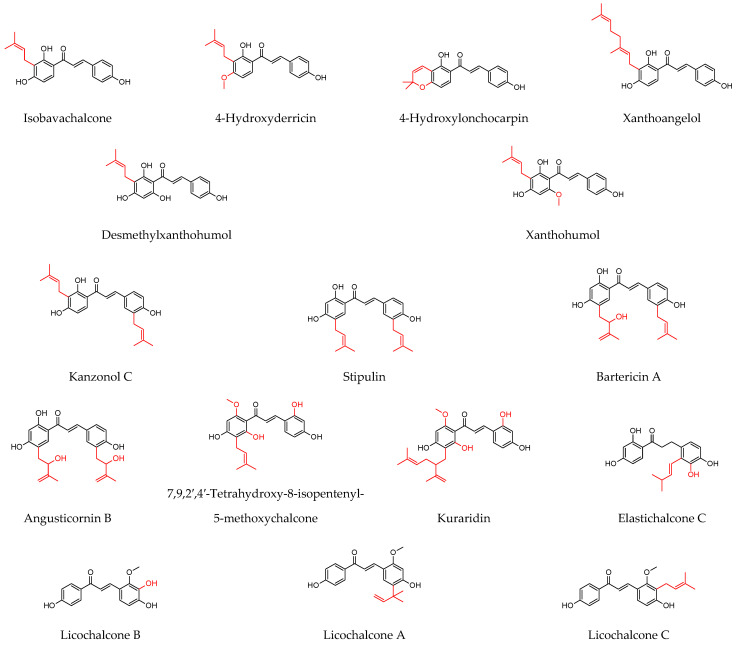
Structures of discussed chalcones.

**Figure 11 molecules-27-04491-f011:**
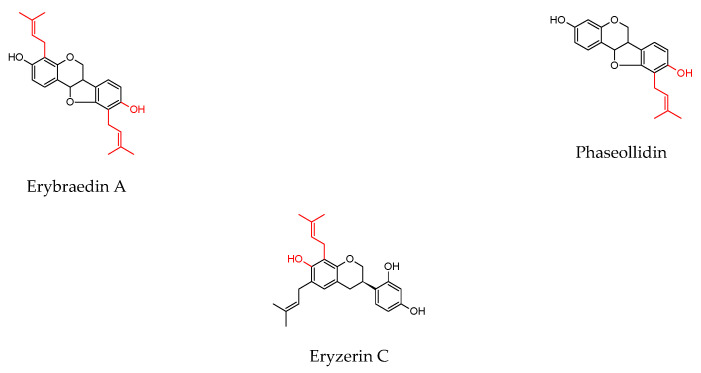
Structures of discussed pterocarpans.

**Figure 12 molecules-27-04491-f012:**
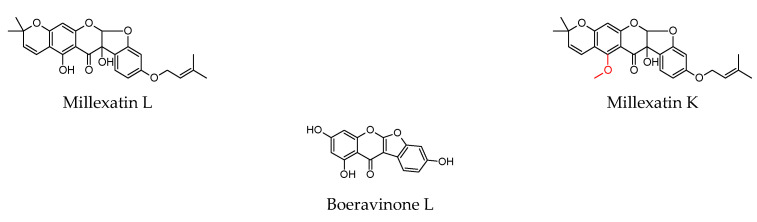
Structures of discussed coumaronochromones.

**Table 1 molecules-27-04491-t001:** Antibacterial activity of known prenylated flavonoids from natural sources.

Plant Source	Compound	Bacteria	IC_50_ (µg/mL) * or MIC (µg/mL)	PC (µg/mL)				Ref.
**Asteraceae**								
** *Helichrysum forskahlii* ** **(J.F. Gmel.) Hilliard and Burtt** **(aerial parts)**	Glabranin	*B. subtilis* *S. aureus*	3 6					[64]
**Celastraceae**								
				AB	CI	VA		[63]
** *Tripterygium wilfordii* ** **Hook.f. (stems and roots)**	Tripteryol B	*C. neoformans* *P. aeruginosa* VRE MRSA	3.0 * 8.6 * 4.3 * 4.5 *	0.8 - - -	- 0.1 - 0.2	- - 3.3 -		
	(±)-5,4′-Dihydroxy-2′-methoxy-6′,6″-dimethypyraro-(2″,3″:7,8)-6-methyflavanone	MRSA *S. aureus*	2.1 * 2.6 *	- -	0.2 0.1	- -		
	((2S)-5,7,4′-Trihydroxy-2′-methoxy-8,5′-di(3-methyl-2-butenyl)-6-methylflavanone	*C. neoformans* MRSA *S. aureus*	1.1 * 2.0 * 2.2 *	0.8 - -	- 0.2 0.10	- - -		
**Euphorbiaceae**								
** *Macaranga tanarius* ** **L. (fruit)**	Propolin D	MSSA (*n* = 2) MRSA *S. epidermidis*	10 * 10 * 10 *					[107]
**Fabaceae**								
				MO				[108]
** *Amorpha fruticosa* ** **L. (fruit)**	Xanthoangelol	*S. aureus* MRSA	25 µM 3.1 µM	<0.9 µM 3.8 µM				
		*S. aureus* MRSA *E. faecalis* VRE *E. faecium* VRE. *faecium* *B. subtilis*	6.3 3.1 12.5 6.3 12.5 3.1 3.1					[109]
				CI				[110]
** *Dalea scandens* ** **(Miller) R. Clausen var. *paucifolia* (roots)**	2(S)-5′-(-1‴,1‴-Dimethylallyl)-8-(3″,3″-dimethylallyl)-2′,4′,5,7-tetrahydroxyflavanone	*S. aureus* MRSA	1.6 1.6	0.2 0.2				
	2(S)-5′-(-1‴,1‴-Dimethylallyl)-8-(3″,3″-dimethylallyl)-2′-methoxy-4′,5,7-trihydroxyflavanone		3.1 3.1					
	5′-(1‴,1‴-Dimethylallyl)-8-(3″,3″-dimethylallyl)-2′,4′,5,7-tetrahydroxyflavone		3.1 3.1					
** *Dalea versicolor* ** **Zucc. var. sessilis (A. Gray) Barneby (whole plants)**	2(S)-5′-(1‴,1‴-Dimethylallyl)-8-(3″,3″-dimethylallyl)-2′,4′,5,7-tetrahydroxyflavanone	*S. aureus*	7.8					[111]
				OX				[112]
** *Dalea searlsiae* ** **(Gray) Barneby** **(roots and aerial parts)**	Malheuran A	OSSA ORSA *B. cereus*	4.3 3.7 3.7	0.4 21.3 106.7				
	Malheuran B		3.4 3.7 2.7					
	Malheuran C		4.6 4.3 3					
	Malheuran D		6.1 6.5 5.0					
	(2S)-5′-(2-Methylbut-3-en-2-yl)-8-(3-methylbut-2-en-1-yl)-5,7,2′,4′-tetrahydroxyflavanone		3.1 3.4 2.3					
	Prostratol F		6.8 6.4 8					
				AP				[113]
** *Derris reticulata* ** **Craib. (stem)**	Lupinifolin	*S. aureus*	8	0.3				
				AP	ER			[114]
** *Echinosophora koreensis* ** **Nakai (roots)**	Kenusanone C	*S. epidermidis* *S. aureus*	20 20	20 1.3	1.3 1.3			
	Isosophoranone		20 20					
	Sophoraisoflavanone A	*E. coli* *S. epidermidis* *S. aureus*	20 20 20	1.3 20 1.3	1.3 1.3 1.3			
	Kenusanone A	*E. coli* *S. epidermidis* *S. aureus*	20 20 20	1.3 20 1.3	1.3 1.3 1.3			
	Sophoraflavanone D		20 20 20					
				NE				[115]
** *Erythrina caffra* ** **Thunb.** **(stem bark)**	Abyssinone V 4′-*O*-methyl ether	*E. coli* *B. subtilis*	3.9 15.6	1.6 0.8				
	6,8-Diprenylgenistein	*S. aureus* *E. coli* *B. subtilis*	7.8 7.8 15.6	0.8 1.6 0.8				
	Alpinumisoflavone		3.9 3.9 7.8					
				CI				[116]
** *Erythrina lysistemon* ** **Hutch. (stem bark)**	Erybraedin A	*B. cereus* *S. aureus* *S. epidermidis* *E. coli*	1 2 2 2	0.02 0.1 0.1 0.1				
	Phaseollidin	*B. cereus* *S. aureus* *S. epidermidis*	10 10 5	0.02 0.1 0.1				
	Abyssinone V 4′-*O*-methyl ether	*B. cereus*	26	0.02				
	Eryzerin C	*B. cereus* *S. aureus* *S. epidermidis* *E. coli* *P. aeruginosa*	10 5 2 5 5	0.02 0.1 0.1 0.1 0.1				
	Alpinumisoflavone	*B. cereus* *S. aureus* *P. aeruginosa*	31 31 20	0.02 0.1 0.1				
	Lysisteisoflavone	*B. cereus* *S. epidermidis* *E. coli* *P. aeruginosa*	2 26 6 31	0.02 0.1 0.1 0.1				
**Larval stage of *Melipotis perpendicularis* (Noctuidae) feeding on the leaves of *Lonchocarpus minimiflorus* Donn. Sm.**	Lonchocarpol A	MRSA VRE. *faecium* *B. megaterium*	0.8–1.6 0.8–1.6 1.0–2.0					[117]
				VA				[118]
** *Millettia extensa* ** **(Benth.) Baker** **(stem)**	Millexatin A	*S. aureus* *S. epidermidis* *B. subtilis*	2 2 2	0.3 0.3 0.3				
	Millexatin F		2 2 2					
	Auriculatin		2 2 2					
	Scandenone		2 4 2					
	Auriculasin		4 4 8					
** *Millettia extensa* ** **(Benth.) Baker** **(leaves and roots)**	Millipurone	*S. epidermidis* *B. cereus* *S. aureus*	4 32 4	0.3 0.1 0.3				[119]
	Millexatin K	*B. cereus* *S. aureus*	32 32	0.1 0.3				
	Millexatin L		32 32					
	Millexatin D		8 8					
	5,7,3′,4′-Tetrahydroxy-6,8-diprenylisoflavone = 6,8-diprenylorobol		16 32					
				CI				[120]
** *Pseudarthria hookeri* ** **Wight and Arn.** **(whole plant)**	Pseudarflavone A	*E. coli* *P. aeruginosa* *S. aureus*	4 16 8	0.3 0.1 0.3				
	6-Prenylpinocembrin	*E. coli* *E. faecalis* *S. aureus*	4 16 8	0.3 8 0.3				
	Boeravinone L	*E. coli*	16	0.3				
** *Psoralea corylifolia* ** **L.** **(seeds)**	Bavachin	*S. aureus* *S. epidermidis*	37 µM 37 µM					[121]
				GE	OF	AB		[122]
** *Retama raetam* ** **Forssk. Webb** **(flowers)**	Licoflavone C	*E. coli* *P. aeruginosa* *C. glabrata* *C. albicans* *C. parapsilosis* *C. krusei*	7.8 15.6 15.6 15.6 15.6 15.6	- 0.5 - - - -	0.1 1 - - - -	- - 0.5 0.5 0.5 0.5		
	Derrone		7.8 15.6 7.8 7.8 7.8 7.8					
				ATB (*n* = 8)				[123]
** *Sophora flavescens* ** **Aiton (roots)**	Kuraridin	*S. aureus* MRSA (*n* = 6)	8 8–16	0.1→128				
				AP	ER			[114]
		*E. coli* *S. epidermidis* *S. aureus*	20 20 20	1.3 20 1.3	1.3 1.3 1.3			
	Kurarinone	*E. coli* *S. epidermidis* *S. aureus*	20 20 20	1.3 20 1.3	1.3 1.3 1.3			
				AP	VA			[124]
		MRSA VRE	2 2	250 250	2.5 150			
	Sophoraflavanone G			AP	ER			[114]
		*E. coli* *S. epidermidis* *S. aureus*	20 20 20	1.3 20 1.3	1.3 1.3 1.3			
				ATB (*n* = 8)				[123]
		*S. aureus* MRSA (*n* = 6)	2 2–4	0.1→128				
				AP	OX			[125]
		MSSA MRSA (*n* = 11)	4 0.5–8	2 64–1024	0.3 256–1024			
	7,9,2′,4′-Tetrahydroxy-8-isopentenyl-5-methoxychalcone	*S. aureus* MRSA (*n* = 5) VRE (*n* = 5)	1.0 1.0–15.6 7.8–15.6					[126]
**Moraceae**								
				VA				[127]
** *Artocarpus elasticus* ** **(** **leaves)**	Elastichalcone C	*S. aureus* MRSA	8 4	1 1				
				OX				[128]
** *Artocarpus integer* ** **(Thunb.) Merr. (roots)**	Artocarpin	*P. acnes* *S. aureus* *S. epidermidis*	2 2 4	0.1 0.5 0.5				
	Cudraflavone C		2 2 4					
				CI				[129]
** *Artocarpus sepicanus* ** **Diels** **(leaves)**	Sepicanin A	MRSA	2.9 µM	0.8				
				AP	ER	AB	MI	[114]
** *Broussonetia papyrifera* ** **(L.) Vent. (root bark)**	Papyriflavonol A	*E. coli* *S. epidermidis* *S. aureus* *S. cerevisiae*	20 10 15 12.5	1.3 20 1.3 -	1.3 1.3 1.3 -	- - - 1.3	- - - 1.3	
	Kazinol B	*S. epidermidis* *S. aureus*	20 20	20 1.3	1.3 1.3	- -		
				GE	NY			[130]
** *Dorstenia angusticornis* ** **Engl.** **(twigs)**	Gancaonin Q	*B. cereus* *B. stearothermophilus* *B. subtilis* *E. faecalis*	0.6 9.8 9.8 0.6	1.2 4.9 1.2 2.4	- - - -			
	Stipulin	*P. aeruginosa* *B. cereus* *E. faecalis*	19.5 4.9 2.4	4.9 1.2 2.4	- - -			
	Angusticornin B	*E. coli* *P. aeruginosa* *B. cereus* *B. megaterium* *B. stearothermophilus* *B. subtilis* *S. aureus* *E. faecalis* *C. albicans* *C. krusei* *C. glabrata*	1.2 9.8 1.2 2.4 2.4 1.2 2.4 1.2 0.6 1.2 0.6	1.2 4.9 1.2 2.4 4.9 1.2 4.9 2.4 - - -	- - - - - - - - 2.4 2.4 9.8			
	Bartericin A	*E. coli* *P. aeruginosa* *B. cereus* *B. megaterium* *B. stearothermophilus* *B. subtilis* *S. aureus* *E. faecalis* *C. albicans* *C. krusei* *C. glabrata*	0.6 <0.3 0.6 1.2 2.4 1.2 0.6 0.6 0.6 1.2 0.6	1.2 4.9 1.2 2.4 4.9 1.2 4.9 2.4 - - -	- - - - - - - - 2.4 2.4 9.8			
				ATB (*n* = 7)				[131]
** *Dorstenia barteri* ** **Bureau var. multiradiata (stems)**	Isobavachalcone	MSSA (*n* = 2) MRSA (*n* = 6)	2–16 4–8	0.02→128				
				GE	NY			[132]
		*E. faecalis* *S. aureus* *B. cereus* *B. megaterium* *B. stearothermophilus* *B. subtilis* *C. albicans* *C. glabrata* *T. rubrum*	0.3 0.3 0.6 0.6 0.3 0.6 0.3 0.3 1.2	4.9 2.4 1.2 2.4 4.9 1.2 - - -	- - - - - 2.4 2.4 4.9 1.2			
	4-Hydroxylonchocarpin	*E. faecalis* *S. aureus* *B. cereus* *B. megaterium* *B. stearothermophilus* *B. subtilis* *C. albicans* *C. glabrata* *T. rubrum*	4.9 4.9 4.9 1.2 1.2 4.9 4.9 4.9 4.9	4.9 2.4 1.2 2.4 4.9 1.2 - - -	- - - - - - 2.4 2.4 1.2			
	Kanzonol C	*E. faecalis* *B. cereus* *B. megaterium* *B. stearothermophilus* *B. subtilis* *C. albicans* *C. glabrata*	4.9 9.8 4.9 4.9 9.8 4.9 4.9	4.9 1.2 2.4 4.9 1.2 - -	- - - - - 2.4 2.4			
				CH	NY			[133]
** *Dorstenia mannii* ** **Hook. f.** **(stems)**	Dorsmanin C	*P. aeruginosa*	4	64	-			
	Dorsmanin E	*C. albicans*	8	-	16			
	Dorsmanin F	*E. coli* *C. albicans*	4 16	2 -	- 16			
	Dorsmanin G	*E. coli* *P. aeruginosa*	16 8	2 64	- -			
				VA	AP			[134]
** *Maclura cochinchinensis* ** **(Lour.) Corner (fruit and leaves)**	Gancaonin M	*E. faecalis* *S. aureus* MRSA *C. albicans*	8 2 2 4	2 0.5 1 0.5	0.5 0.3 0.5 0.3			
	Lupiwighteone		8 4 8 8					
	Lupalbigenin		2 1 1 4					
	Scandenone		4 2 2 8					
	Auriculatin		2 2 2 2					
	Millexatin F		4 1 2 4					
	Derrone	*S. aureus* MRSA *C. albicans*	4 4 32	0.5 1 0.5	0.3 0.5 0.3			
	Macluracochinone E		8 32 32					
				AP	OF	KE		[135]
** *Maclura pomifera* ** **(Rafin.) Schneider (fruit)**	Scandenone	*E. coli* *S. aureus* *B. subtilis* *E. faecalis* *C. albicans*	2 0.5 8 0.5 1	2 0.1 0.5 0.5 -	0.1 0.5 1 1 -	- - - - 1		
** *Morus alba* ** **L. (root bark)**				KS	AP	ME		[136]
	Kuwanon C	*S. aureus* MRSA (*n* = 3)	2 2–4	2 4	4 8–16	32 128–256		
				AP	CI	VA		[137]
		MRSA (*n* = 3) *E. faecalis* VRE (*n* = 3)	2 4 4–8	>16 8 8	8–128 <8 NT	- <16 512→>1024		
				VA				[138]
		MSSA MRSA *B. subtilis* *E. faecalis*	2 2 4 4	2 2 ≤0.3 >128				
				AP	CI	VA		[137]
	Kuwanon T	MRSA (*n* = 3) *E. faecalis*	4 4	>16 8	8–128 <8	- <16		
	Morusinol	MRSA MRSA (*n* = 2)	16	>16	16	-		
	Kuwanon U		4–8	>16	16–128	-		
				KS	AP	ME		[136]
		*S. aureus* MRSA (*n* = 3)	4 4–8	2 4	4 8–16	32 128–256		
				AP	CI	VA		[137]
	Kuwanon E	MRSA (*n* = 3)	4–8	>16	8–128	-		
				KS	AP	ME		[136]
		*S. aureus* MRSA (*n* = 3)	4 4	2 4	4 8–16	32 128–256		
				ATB (*n* = 6)				[139]
		MSSA MRSA (*n* = 10)	4 4–16	4–128 4–256				
				AP	CI	VA		[137]
	Morusin	MRSA (*n* = 3) *E. faecalis* VRE (*n* = 3)	2–4 8 4–8	>16 8 8	8–128 <8 -	- <16 512→>1024		
				KS	AP	ME		[136]
		*S. aureus* MRSA (*n* = 3)	4 2–8	2 4	4 8–16	32 128–256		
				VA				[138]
		MSSA MRSA *B. subtilis* *E. faecalis*	8 8 4 8	2 2 ≤0.3 >128				
				ATB (*n* = 6)				[139]
		MSSA MRSA (*n* = 10)	16 8–32	4–128 4–256				
				KS	AP	ME		[136]
	5′-Geranyl-5, 7, 2′, 4′ -tetrahydroxyflavone	*S. aureus* MRSA (*n* = 3)	2 2–4	2 4	4 8–16	32 128–256		
	Kuwanon B		4 4					
** *Morus mongolica* ** **Schneider (root bark)**				AP	ER			[114]
	Morusin	*S. epidermidis*	20	20	1.3			
	Kuwanon C	*E. coli* *S. epidermidis* *S. aureus*	10 6.3 6.3	1.3 20 1.3	1.3 1.3 1.3			
**Paulowniaceae**								
** *Paulownia tomentosa* ** **(Thunb.) Steud** **(fruit)**	Tomentodiplacone B			ATB (*n* = 8)				[98]
		*S. aureus* MRSA (*n* = 5)	8 8–16	0.1→32				
	Mimulone		2 2–4					
				CI				[140]
		*B. cereus* *B. subtilis* *E. faecalis* *S. aureus*	4 4 4 8	1 2 1 0.5				
	Diplacone		4 4 4 4					
				ATB (*n* = 8)				[98]
		*S. aureus* MRSA (*n* = 5)	8 8–16	0.1→32				
	3′-*O*-methyl-5′-hydroxydiplacone		4 4–8					
				CI				[140]
		*B. cereus* *B. subtilis* *E. faecalis* *S. aureus*	4 4 4 2 4	1 2 1 0.5				
	3′-*O*-methyl-5′-*O*-methyldiplacone		4 4 4 4					
				ATB (*n* = 8)				[98]
		*S. aureus* MRSA (*n* = 5)	2 4–8	0.1→32				
	3′-*O*-methyldiplacol		4 4–8					
				CI				[140]
		*B. cereus* *B. subtilis* *E. faecalis* *S. aureus*	2 4 4 2	1 2 1 0.5				
	3′-*O*-methyldiplacone		4 8 8 8					
	Tomentodiplacone	*B. cereus* *S. aureus*	16 16	1 0.5				
**Propolis**								
**Taiwanese green propolis**	Propolin C	*B. subtilis* *S. aureus* (*n* = 3)	2.5 1.3–5					[141]
	Propolin D		5 10–20					
	Propolin F		10 10–20					
	Propolin G		10 10–20					
**Prenylated flavonoids obtained from indefinite source**								
**FLAVONES**								
**Albanin A**		MSSA MRSA	32 32					[142]
**Artocarpin**			1 2					
**Broussoflavonol F**			16 16					
**Corylifol C**		MSSA	16					
**Glyasperin A**				OX	VA	CH	ST	[143]
		*S. aureus*	16–32	0.3–0.5	0.5–1	4	8–16	
**Kuwanon A**		MSSA MRSA	4 8					[142]
**Kuwanon C**			1 1					
**Licoflavone B**			32 32					
			16 32					[138]
**Licoflavone C**		MSSA	32 32					[142,144]
**Licoflavonol**		MSSA MRSA	8 8					[142]
**Morusin**			8 8					
**3′-Geranyl -3-prenyl-5,7,2′,4′-tetrahydroxyflavone**			4 4					
**5,7,2′,4′-Tetra-hydroxy-3-geranylflavone**			8 8					
**5,7,4′-Trihydroxy-3,6-dimethoxy-3′,5′-diprenylflavone**		MRSA	32					
**5′-Geranyl-5,7,2′,4′-tetrahydroxyflavone**		MSSA MRSA	2 2					
**6-Prenylapigenin**			32 32					
**6-Prenylquercetin 3-methyl ether**		MSSA	32					
**8-Prenylkaempferol**		MSSA MRSA	32 32					
**ISOFLAVONES**								
**Eurycarpin A**		MSSA MRSA	8 8					[142]
**Gancaonin M**			8 8					
**Glycyrrhisoflavone**			32 32					[144]
**Isoneobavaisoflavone**			4 4					[142]
**Isowighteone**			16 16					
**Licoisoflavone A**			32 32					[142,144]
**Licoisoflavone B**				OX	VA	CH	ST	[143]
		*S. aureus*	4–8	0.3–0.5	0.5–1	4	8–16	
**Lupalbigenin**		MSSA MRSA	1 1					[142]
**Lupiwighteone**		MSSA	32					
**Scandenone, syn. Warangalone**		MSSA MRSA	8 2					
**Sophoraisoflavone A**			32 32					
**Wighteone**			8 8					
**6,8-Diprenylgenistein**			4 2					
**6,8-Diprenylorobol**			16 8					
**8-Prenyldaidzein**			32 32					
**8-Prenylluteone**			8 8					
**FLAVANONES**								
**Bavachin**		MSSA	32					[142]
**Bavachinin A**		MSSA MRSA	4 8					
**Euchrestaflavanone A**			2 2					
**Glabrol**			1 1					
			2 2					[144]
				VA				
		MRSA (*n* = 20)	1–4	2–16				
**Leachianone G**		MSSA MRSA	32 32					[142]
**Licoflavanone**			32 32					
**Sophoraflavanone C**			2 2					
**6-Prenylnaringenin**			8 8					
**CHALCONES**								
**Desmethylxanthohumol**		MSSA MRSA	16 16					
**Isobavachalcone**		MSSA MRSA	4 4					
				TE				[145]
			1.6 3.1	0.2 >5.9				
**Kanzonol C**			4 4					[142]
**Licochalcone A**			4 4					
			2 4					[144]
				VA				[144]
		MRSA (*n* = 20)	1–8	2–16				
				OX	VA	CH	ST	[143]
		*S. aureus*	4	0.3–0.5	0.5–1	4	8–16	
**Licochalcone B**		MRSA	16					[142,144]
**Licochalcone C**		MSSA MRSA	4 4					
				VA				[144]
		MRSA (*n* = 20)	1–16	2–16				
**Licochalcone D**		MSSA MRSA	16 32					[142]
			32 16					[144]
**Licochalcone E**			4 4					[142]
			4 4					[144]
				VA				[144]
		MRSA (*n* = 20)	0.5–16	2–16				
				OX				[146]
		*S. aureus* MRSA (*n* = 6)	2 1–4	0.3 0.1–256				
**Xanthoangelol**		MSSA	32					[142]
**Xanthohumol**		*S. aureus* (*n* = 3)	15.6–62.5					[147]
		MSSA MRSA	4 4					[142]
		*S. epidermidis* (*n* = 2) *S. capitis* ssp. *ureolyticus* *S. aureus* *MRSA*	2 2 2 4					[148]
**4-Hydroxyderricin**		MSSA MRSA	2 2					[142]
**ISOFLAVEN**								
**Glabrene**		MSSA MRSA	16 16					[144]

**Abbreviations:** IC_50_ (half-maximal inhibitory concentration), MIC (minimum inhibitory concentration), NT (not tested), PC (positive control). **Microorganisms tested for the antimicrobial activity:**
*B. cereus* (*Bacillus cereus*), *B. megaterium* (*Bacillus megaterium*), *B. stearothermophilus* (*Bacillus stearothermophilus*), *B. subtilis* (*Bacillus subtilis*), *C. albicans* (*Candida albicans*)*, C. glabrata* (*Candida glabrata*), *C. krusei* (*Candida krusei*), *C. neoformans* (*Cryptococcus neoformans*), *E. coli* (*Escherichia coli*), *E. faecalis* (*Enterococcus faecalis*), MRSA (methicillin-resistant *Staphylococcus aureus*), MSSA (methicillin-sensitive *Staphylococcus aureus*), ORSA (oxacillin-resistant *S. aureus*), OSSA (oxacillin-sensitive *S. aureus*), *P*. *aeruginosa* (*Pseudomonas aeruginosa*), *S. aureus* (*Staphylococcus aureus*), *S. capitis* spp. *ureolyticus* (*Staphylococcus capitis* spp. *ureolyticus*), *S. cerevisiae* (*Saccharomyces cerevisiae*), *S. epidermidis* (*Staphylococcus epidermidis*), *T. rubrum* (*Trichophyton rubrum*), VRE (vancomycin-resistant *Enterococcus*), VRE. *faecium* (vancomycin-resistant *Enterococcus faecium*). **Standard antibiotics and antifungals:** AB (amphotericin B), AM (amoxicillin), AP (ampicillin), CE (cefazolin), CH (chloramphenicol), CI (ciprofloxacin), CL (clarithromycin), CO (colistin), ER (erythromycin), GE (gentamicin), KE (ketoconazole), KS (kanamycin sulphate), ME (methicillin), MI (miconazole), MO (moxifloxacin), NE (neomycin), NY (nystatin), OF (ofloxacin), OX (oxacillin), PG (penicillin G), RI (rifampicin), ST (streptomycin), TE (tetracycline), VA (vancomycin).

**Table 2 molecules-27-04491-t002:** Multiple beneficial activities of natural compounds involved in wound healing.

Compound	Cytotoxic Activity	Anti-Inflammatory Activity	Antioxidant Activity	↓ Bacterial Pathogenicity
**Alpinumisoflavone**	Weak cytotoxicity in PC-3 cells [149]	5, 10 µg/mL → ↓ TNF-α, IL-6, IL-1β, IL-17, ICAM-1, NO in LPS-stimulated RAW 264.7 cells [150].	DPPH scavenging activity, IC_50_ = 54.0 µg/mL [151].	
		1, 5, 10 mg/kg i.p 1 h before → protective effect against pulmonary inflammation in LPS-stimulated acute lung injury in mice [150].	5, 10 µg/mL → ↑ the levels of CAT, HO-1, GPx, SOD in LPS-stimulated RAW 264.7 cells [150].	
		25 and 50 µM → inhibition of TNF-α-induced ↑ in MMP-1, ↓: procollagen I α1, NOS, COX-2, IL-1β, IL-6, IL-8, NF-κB, MAPKs [152].	↓ ROS and NO in TNF-α-treated HDFs [152].	
**Artocarpin**	IC_50_ = 45.3 μM in RAW 264.7 cells [153].	Inhibition of LPS-induced NO production in RAW 264.7 cells, IC_50_ = 18.7 µM [153].	TEAC_ABTS_ = 0.9 mM [154].	Synergy with norfloxacin against MRSA, *P. aeruginosa*, and *E. coli*. Synergy with tetracycline against MRSA and *P. aeruginosa*. Synergy with ampicillin against MRSA [155].
	IC_50_ = 7.9 µM in PC-3 cells. IC_50_ = 8.3 µM in NCI-H460 cells [156].			
	ED_50_ = 3.3 µg/mL in MCF-7 cells. ED_50_ = 3.8 µg/mL in MDA-MB-231 cells. ED_50_ = 3.3 µg/mL in A549 cells. ED_50_ = 3.4 µg/mL in 1A9 cells. ED_50_ = 3.8 µg/mL in HCT-8 cells. ED_50_ = 4.9 µg/mL in CAKI-1 cells. ED_50_ = 5.4 µg/mL in SK-MEL-2 cells. ED_50_ = 3.7 µg/mL in U87-MG cells. ED_50_ = 4.1 µg/mL in PC-3 cells. ED_50_ = 3.2 µg/mL in KB cells. ED_50_ = 3.6 µg/mL in KB-VIN cells. [157].	Topical dose 0.05–0.1% ↓ TNF-α levels, COX-2 and cPLA2 protein expressions in the skin homogenate. Photoprotective effect on ultraviolet B (UVB)-induced skin damage in hairless mice [158].	0.05% artocarpin treatment prevents UVB-induced oxidative stress by affecting antioxidant activity [158].	
	IC_50_ = 5.1 μmol/L in PC-3 cells. IC_50_ = 10.2 μmol/L in NCI-H460 cells. IC_50_ = 8.1 μmol/L in A-549 cells [159].			
	IC_50_ = 5.1 µg/mL in KB cells. IC_50_ = 3.3 µg/mL in BC cells. IC_50_ = 5.6 µg/mL in Vero cells [160].			
**Bavachin**	CC_50_ = 20.2 µM in Hep3B cells [161].	Inhibitory effect on IL-6-induced STAT3 promoter activity in Hep3B cells, IC_50_ = 4.9 µM [161].		
		Suppression of LPS-induced NO and PGE_2_ production, and ↓ iNOS and mPGES-1 expression. ↓ of LPS-induced IL-6 and IL-12p40 production and ↓ the activation of MAPKs and NF-κB. Suppression of NLRP3 inflammasome-derived IL-1β secretion, ↓ caspase-1 activation, repression of mature IL-1β expression, and inhibition of inflammasome complex formation [162].		
		Downregulation of IL-4 in the spleen of T cells from 4get IL-4-GFP mice. ↓ the IL-4 levels by downregulating the level of *Gata-3* expression and STAT6 phosphorylation. 50 mg/kg dissolved in the solution by daily lavage administration [163].		
**Diplacone = propolin C = nymphaeol A**	IC_50_ = 14.3 µM in WB-F344 cells [164].	10 µM ↓ the expression of TNF-α and MCP-1 and ↑ the expression of ZFP36 [165].	DPPH scavenging by SC_50_ = 3.2 µg/mL) [166].	Dose-dependent inhibition of *S. aureus* biofilm formation [107].
		Inhibition of IκB-α degradation, ↓ of COX-2 expression [167].		
		COX-1 inhibitor IC_50_ = 1.8 μM COX-2 inhibitor IC_50_ = 4.2 μM 5-LOX inhibitor IC_50_ = 0.1 μM [168].		
	Antiproliferative (IC_50_ = 9.3 μM) and cytotoxic (LC_50_ = 18.0 µM) effect in THP-1 cells [169].	25 mg/kg prior and after induction of colitis ameliorates its symptoms and delays the onset. ↓ of the levels of COX-2 and ↑ the ratio of pro-MMP2/MMP2 activity, ↓ of SOD2 and CAT [170].	TEAC_ABTS_ 3.2, TEAC_DPPH_ = 1.1, TEAC_FRAP_ = 0.5, TEAC_Inhibition of. peroxynitrite induced tyrosine nitration_ *=* 0.8. Superoxide scavenging activity-enzymatic = 45.2%, nonenzymatic = 25.9% at 50 μM [171].	
	EC_50_ =< 10 µM in MCF-7 cells. EC_50_ = 3.2 µM in CEM cells. EC_50_ =< 10 µM in RPMI8226 cells. EC_50_ = 2.4 µM in U266 cells. EC_50_ =< 10 µM in HeLa cells. EC_50_ = 5.9 µM in BJ cells. EC_50_ =< 10 µM in THP-1 cells [172].	Inhibition of LPS-induced NO production in RAW 264.7 cells, IC_50_ = 5.0 µM [173].	DPPH quenching activity TEAC 5.2 at 10 µM [164].	
	At tested concentrations did not inhibit cell proliferation; it induced cell proliferation to some extent in RAW 264.7 cells [174].	Inhibition of albumin denaturation, IC_50_ = 0.3 µM. Inhibition of nitrite production stimulated by LPS in RAW 264.7 cells, IC_50_ = 3.2 µM. COX-2 inhibitor = 11.7 µM [174].	DPPH radical scavenging activity, IC_50_ = 6.5 µg/mL [175].	
**Glabrene**	Cytotoxic activity for: HepG2 cells (10 μM) = 25.9%. SW480 cells (10 μM) = 30.7%. A549 cells (10 μM) = 0%. MCF7 cells (10 μM) = 17.6% [176].	10 μM inhibited LPS-induced NO production in RAW 264.7 cells by 57.5%, IC_50_ = 9.5 μM. 10 μM inhibited LPS-induced NF-κB activation by 41.7% [176].	10 μM treated HepG2 cells transfected with the ARE luciferase reporter gene (HepG2C8 cells) to evaluate Nrf2 activation. 2.7-fold of control for Nrf2 activation activity [176].	
**Isobavachalcone**	IC_50_ = 2.90 μM (CCRF-CEM cells) to >123. 46 μM (AML12 cells) [177].	20 μg/mL and 50 μg/mL → suppression of iNOS expression induced by TLR agonists in murine macrophages [178].	Peroxyl radical scavenging activity with an ORAC value of 24.8 μM [179].	Antibiofilm activity with MBIC = 0.8 µg/mL against MSSA and MRSA → 75% inhibition of biofilm formation [145].
	Cell viability = 98.2% at 50 μM, 64.8% at 100 μM in RAW 264.7 cells [178].	Inhibition of NO production in LPS-activated RAW 264.7 cells, IC_50_ = 6.4 µM [153].		
	IC_50_ = 16.4 µM in RAW 264.7 cells [153].			
	IC_50_ =< 20 µM in NB4, U937, K562s, K562r cells. IC_50_ = >20 µM in HL60, THP-1, U937, MOLM-13 cells. IC_50_ = 75.5 µM in HCT116 cells. IC_50_ = 44.1 µM in SW480 cells. IC_50_ = 128.3 µM in Tca8113 cells. IC_50_ = 16.5 µM in HepG2 cells. IC_50_ = 13.2 µM in Hep3B cells. IC_50_ =< 40 µM in MCF-7, ZR-75–1, MDA-MB-231 cells. IC_50_ = 26.2 µM in PC-3 cells. IC_50_ = >50 µM in LNCaP cells. IC_50_ = 15.1 µM in PC-3 cells. IC_50_ = >50 µM in HeLa cells [180].	Inhibition of NO production in LPS-activated RAW 264.7 cells, IC_50_ = 17 µM [181].	Inhibition of NADPH-, ascorbate-, t-BuOOH-, and CCl_4_-induced lipid peroxidation in microsomes, IC_50_ = 57.3, 20.8, 61.7, 17.6 μM, respectively [182].	
		3.12 µg/mL inhibited NO production by 79.57% in LPS-activated RAW 264.7 cells. 15-LOX inhibitor, IC_50_ = 25.9 µg/mL [183].		
		Attenuated Sephadex-induced lung injury in rats, inhibition of NF-κB-mediated upregulation of A20 and activation of NRF2/HO-1 signalling pathway [184].		
	IC_50_ = 31.6 µM in L-02 IC_50_ = 31.3 µM in HUVEC [185].			
	IC_50_ = >100 µM in cerebellar granule cells [186].			
	↓ of the cell viability of HaCaT cells at 25 µg/mL after 24 h [145].	Inhibitory effect on IL-6-induced STAT3 promoter activity in Hep3B cells, IC_50_ = 2.5 µM [161].		
**Isosophoranone**	CC_50_ in the range of 25–62 µM in human tumour cells (HSC-2, HSG) and human normal cells (HGF, HPC, HPLF) [187].	Inhibition of NO production in LPS-activated RAW 264.7 cells, (IC_50_ = 17 µM) [187].		
**Kazinol B**	**(2*S*)-Kazinol B** IC_50_ = >100 µM in Bcap37, MCF-7, U251, A549 cells. IC_50_ = 58.4 µM in HepG2 cells. IC_50_ = 38.9 µM in Hep3B cells. **(2*R*)-Kazinol B** IC_50_ = >100 µM in Bcap37, MCF-7, U251, A549 cells. IC_50_ = 64.2 µM in HepG2 cells. IC_50_ = 30.3 µM in Hep3B cells [188].	Inhibition of NO production in LPS-activated RAW 264.7 cells (IC_50_ = 21.6 µM) via inhibition of iNOS activity [189].	Protection of mitochondria from injury through direct Fyn inhibition [190].	
**Kuraridin**	Noncytotoxic when compared with the drug-free control in the range of 0.3–64 µg/mL in PBMC cells [123].	COX-1 inhibitor IC_50_ = 0.6–1 µM. 5-LOX inhibitor IC_50_ = 5.4–6.9 µM [191].		Additive effect with ciprofloxacin, erythromycin, gentamicin, kanamycin, oxacillin [123].
	IC_50_ = 37.8 μg/mL in HepG2 cells [114].			
**Kurarinone**	Inhibition of fatty acid β-oxidation through the reduction of l-carnitine and the inhibition of the PPAR-α pathway → lipid accumulation and liver injury (hepatotoxicity) [192].	COX-1 inhibitor IC_50_ = 0.6–1 µM. 5-LOX inhibitor IC_50_ = 22 µM [191].	Activation of Nrf2 and ↑ expression of antioxidant enzymes, including HO-1 [193].	
	Little toxic effects in BEAS-2B. In vivo apparent signs of toxicity [194].	Inhibition of the expression of interleukin IL-1β, iNOS in LPS-stimulated RAW 264.7 cells [193].		
	IC_50_ = 2–62 µM in cervical, lung (non-small and small), hepatic, esophageal, breast, gastric, cervical, and prostate cancer cells 20–500 mg/kg *in vivo* in lungs (non-small and small) cancer. Higher selectivity toward cancer cells in comparison with respective normal cells [195].	Psoriasis-like skin disease induced by IL-23 and contact dermatitis induced by TNCB. Repression of disease development by inhibiting the expression of proinflammatory mediators and through the suppression of pathogenic CD4+T-cell differentiation and the overall immune response [196].		
		Inhibition of LPS-induced macrophage activation and expression of proinflammatory genes, while ↑ anti-inflammatory gene expression including IL-10 in an AhR-dependent manner. An immunomodulatory activity in the treatment of IBS [197].		
**Kuwanon A**		Inhibition of NO production stimulated by LPS and IFN-γ in RAW 264.7 cells, IC_50_ = 10.5 μM [198].		
		COX-2 inhibitor IC_50_ = 14 μM [199].		
**Kuwanon C**	IC_50_ = 14.2 µM in B16 melanoma cells [200].	Inhibition of NO production stimulated by LPS and IFN-γ in RAW 264.7 cells, IC_50_ = 12.6 μM [198].		
	IC_50_ = 1.7 µM in THP-1 cells [201].	5-LOX inhibitor IC_50_ = 12 µM. 12-LOX inhibitor IC_50_ = 19 µM [191].		
	IC_50_ = 3.9 μM in MCF-7 cells IC_50_ = 9.54 μM in HepG2 cells [202].	Anti-inflammatory effects of kuwanon C are regulated by HO-1 expression [203].		
**Kuwanon E**	Noncytotoxic EC_50_ > 10 µM in MCF-7, CEM, RPMI8226, U266, HeLa, BJ, THP-1 cells [172].	Inhibition of NO production stimulated by LPS and IFN-γ in RAW 264.7 cells, IC_50_ = 14.9 μM [198].		Synergy with amikacin and etimicin [139].
		Inhibition of IL-6 production, IC_50_ = 47.5 μM without a cytotoxic effect in A549 cells [204].		
		COX-2 inhibitor IC_50_ = 34 µM [199].		
**Kuwanon G**	Toxic effect in RAW 264.7 cells ≥ 50 μM. The viability of cells was not affected at concentrations of 2, 5, 10, and 20 μM [205].	Inhibition of NO production at 100 μM (79.9%) [204].		
		↓ of the release of RANTES/CCL5, TARC/CCL17, and MDC/CCL22 via downregulation of STAT1 and NF-κB p65 signalling in TNF-α- and IFN-γ-stimulated HaCaT keratinocytes. Inhibition of histamine production and 5-LOX activation in PMA- and A23187-stimulated MC/9 mast cells [206].		
		20 µM ↓ the ox-LDL induced inflammatory response by suppressing the NF-κB activation in RAW 264.7 cells [205].		
**Licochalcone A**	Cytotoxic activity for: HepG2 cells (10 μM) = 14%. SW480 cells (10 μM) = 7%. MCF7 cells (10 μM) = 10% [207].	Inhibition of NF-κB transcription, IC_50_ = 13.9 μM [207].	Inhibition of peroxyl radical-induced DCFH oxidation without a PBS wash (EC_50_ = 58.8 μg/mL) and with a PBS wash (EC_50_ = 46.3 μg/mL) [208].	Subinhibitory concentrations ↓ the secretion of SEA and SEB by both MSSA and MRSA [209].
	IC_50_ for 24 and 48 h = 6.0 and 13.7 μg/mL, respectively, for the HepG2 cells [208].	Inhibitor for *P. acnes*-induced NLRP3 inflammasome activation. Block of *C. acnes*-induced production of caspase-1 (p10) and IL-1β in macrophages and SZ95. Suppression of *C. acnes*-induced ASC speck formation and mitochondrial reactive oxygen species [210].		
	IC_50_ = 4.8 μM in A549 cells. IC_50_ = 4.6 μM in SK-OV-3 cells. IC_50_ = 2.7 μM in SK-MEL-2 cells. IC_50_ = 3.4 μM in HCT-15 cells [211].	Suppression of ORAI1, Kv1.3, and KCa3.1 channels, IC_50_ = 3, 0.8, and 11.2 µM, respectively. Suppressive effects on the IL-2 secretion and proliferation in CD3 and CD28 antibody-induced T-cells [212].		
	IC_50_ = 36.6 µg/mL in HepG2 cells. IC_50_ = 26.9 µg/mL in Vero cells [144].	Inhibition of LPS-induced phosphorylation at serine 276 and transcriptional activation of NF-κB. Inhibition of LPS-induced activation of PKA [213].	↑ protein expression of SOD1, CAT, and GPx1 in a concentration-dependent manner (2–8 μg/mL for 24 h) [208].	
		Attenuation of LPS-induced kidney histopathologic changes, serum BUN, and creatinine levels. Suppression of LPS-induced TNF-α, IL-6, and IL-1β production in both serum and kidney tissues. Inhibition of LPS-induced NF-κB activation [214].		
		Inhibition of PGE2 and NO production and iNOS and COX-2 expression, induced by IL-1β. Inhibition of MMP-1, MMP-3, and MMP-13 production in IL-1β-stimulated chondrocytes. Inhibition of phosphorylation of NF-κB p65 and IκBα. Upregulation of the expression of Nrf2 and HO-1 [215].		
		Inhibition of sUV-induced COX-2 expression and PGE_2_ generation through the inhibition of AP-1 transcriptional activity. Suppression of sUV-induced phosphorylation of Akt/mTOR and ERK1/2/p90 ribosomal protein S6 kinase in HaCaT cells. Suppression of the activity of PI3K, (MEK)1, and B-Raf, but not Raf-1 in cell-free assays [216].		
**Licochalcone B**		Inhibition of LPS-induced phosphorylation at serine 276 and transcriptional activation of NF-kappaB. Inhibition of LPS-induced activation of PKA. Reduction of the LPS-induced production of NO, TNFα, and MCP-1 [217].	Suppression of the oxidative stress and inflammation, manifesting as the enhancement of SOD, GSH, and IL-4, but the decline of MDA, iNOS, and TNF-α [218].	
		A specific inhibitor of the activation of the NLRP3 inflammasome in macrophages, no effect on the activation of AIM2 or NLRC4 inflammasome. It binds to NEK7 and inhibits the interaction between NLRP3 and NEK7 → suppressing NLRP3 inflammasome activation. Protective effects in mouse models of NLRP3 inflammasome-mediated diseases → LPS-induced septic shock, MSU-induced peritonitis, and nonalcoholic steatohepatitis [219].		
**Licochalcone C**		↓ NF-κB translocation and several downstream molecules, -iNOS, ICAM-1, and VCAM-1. Upregulation of the PI3K/Akt/eNOS signalling pathway [220].	50 µM attenuates inflammatory response by influencing extracellular O_2_^−^ production and by modulating the antioxidant network activity of SOD, CAT, and GPx activity [221].	
		Attenuation of the LPS-IFN-γ-induced inflammatory response by ↓ the expression and activity of iNOS via NF-κB [221].		
**Licochalcone E**	IC_50_ = 5.9 μM in A549 cells. IC_50_ = 5.2 μM in SK-OV-3 cells. IC_50_ = 2.9 μM in SK-MEL-2 cells. IC_50_ = 3.4 μM in HCT-15 cells [211].	Dose-dependent inhibition of IL-12p40 production from LPS-stimulated RAW 264.7 cells. ↓ binding to the NF-κB site in RAW 264.7. Inhibition of the ↑ IL-12p40 expression and ear thickness induced by oxazolone in chronic allergic contact dermatitis model [222].		Subinhibitory concentrations → a dose-dependent decrease in α-toxin expression in *S. aureus* [146].
		Topical application of 0.5–2 mg inhibited TPA-induced ear oedema formation; phosphorylation of SAPK/JNK, c-Jun, and extracellular signal regulated kinase ½, and expression of iNOS and COX-2 in mouse skin. 2.5–7.5 μmol/L → ↓ in LPS-induced release of NO and PGE_2_; ↓ mRNA expression and secretion of IL-6, IL-1β, and TNF-α; ↓ promoter activity of iNOS and COX-2 and expression of their corresponding mRNAs and proteins; ↓ activation of AKT, MAPK, SAPK/JNK, and c-Jun; ↓ phosphorylation of IκB kinase-αβ and IκBα, degradation of IκBα, translocation of p65 to the nucleus and transcriptional activity of NF-κB; and transcriptional activity of AP-1 in RAW 264.7 cells [223].		
**Licoflavanone**		Inhibition of NO production in LPS-stimulated RAW 264.7 cells, IC_50_ = 37.7 µM [224].	IC_50_ = 59.6 µM in ABTS assay [224].	
		↓ of NF-kB translocation into the nucleus→ ↓ proinflammatory cytokines and COX-2/iNOS expression levels. ↓ of p38, JNK, and ERK1/2 phosphorylation and activation. Disruption of the NF-kB/MAPKs signal transduction pathway → ↓ in mRNA levels of TNFα, IL 1β, and IL 6 [224].		
**Licoflavone C =** **8-prenylapigenin**	IC_50_ = 9 µg/mL in Hep-2 cells [122].	Inhibition of the LPS-induced gene expression for TNF-α, iNOS, COX-2, and release of TNF-α, NO, and PGE_2_, through the inhibition of NF-κB activation and reactive oxygen species accumulation in RAW 264.7 cells [225].	↓ of increase in the cellular ROS levels at 3 μM in RAW 264.7 cells [225].	
	IC_50_ = 121.4 μM in RAW 264.7 cells [225].			
	IC_50_ = 41.6 μM in RAW 264.7 cells [153].			
	IC_50_ = 42 μmol/L in H4IIE cells. IC_50_ = 37 μmol/L in C6 glioma cells [226].			
		Inhibition of NO production in LPS-activated RAW 264.7 cells, IC_50_ = 20.4 µM [153].		
**Lonchocarpol A**		COX-1 inhibitor IC_50_ = 16.9 μM. COX-2 inhibitor IC_50_ = 9.5 μM [227].		
		↓ of NO production, IC_50_ = 2.5 μM and ↓ NOX activity, IC_50_ = 24.4 μM in murine microglial cells [228].		
**Lupalbigenin**	IC_50_ = 11.630–37.712 µM in MCF-7, MDA-MB-231, MDA-MB-468, SW-620, and the mouse fibroblast cell line L-929, [229].	1.25 and 2.5 mM effectively inhibited the LPS-induced TNF-α, COX-2, iNOS, and NF-κB [230].		
**Mimulone**	Cytotoxicity < 50% of DMSO in WB-F344 [164].	COX-1 inhibitor IC_50_ = 3.6 μM. COX-2 inhibitor IC_50_ = 6.0 μM [168].	TEAC_ABTS_ = 1.7 [171].	Synergy with oxacillin, additive effect with tetracycline and ciprofloxacin [231].
	IC_50_ = 6.6 µM in THP-1 cells [232].	25 mg/kg prior and after induction of colitis ameliorated its symptoms and delayed the onset. ↓ of the levels of COX-2 and ↑ the ratio of pro-MMP2/MMP2 activity, ↓ of SOD2, and CAT [170].	DPPH quenching activity TEAC 0.4 at 10 µM [164].	
**Morusin**	IC_50_ = 0.6 µM in HeLa. IC_50_ = 7.9 µM in MCF-7. IC_50_ = 9.2 µM in Hep3B [233].	Inhibition of NO production stimulated by LPS and IFN-γ in RAW 264.7 cells, IC_50_ = 10.6 μM [198].		
		Inhibition of RANTES/CCL5 and TARC/CCL17 secretion via the suppression of STAT1 and NF-κB p65 phosphorylation in TNF-α- and IFN-γ-stimulated HaCaT keratinocytes, and the release of histamine and LTC_4_ by suppressing 5-LOX activation in PMA- and A23187-stimulated MC/9 mast cells [206].		
**Morusinol**	IC_50_ = 4.3 µM in THP-1 cells [201].	The attenuation of LPS-induced secretion of TNF-α → an effect nearly twice that of prednisone [201].		Synergy with amikacin, ciprofloxacin, vancomycin, streptomycin [139].
**Papyriflavonol A**	IC_50_ = 20.9 µg/mL in HepG2 cells [114].	5-LOX inhibitor IC_50_ = 7 µM [191].		
		Inhibition of sPLA2s-IIA (IC_50_ = 3.9 µM) and -V (IC_50_ = 4.5 µM). Inhibition of LTC_4_ (IC_50_ = 0.6 µM) production in mouse bone marrow mast cells. 12.5–50 mg/kg i.p. reduced IgE-dependent PCA [234].		
**Propolin D** **= nymphaeol B**	12 µM inhibited cell proliferation in RAW 264.7 cells [174].	Inhibition of albumin denaturation, IC_50_ = 0.5 µM. Inhibition of nitrite production stimulated by LPS in RAW 264.7 cells, IC_50_ = 5.4 µM. COX-2 inhibitor = 17.9 µM [174].	DPPH radical scavenging activity, IC_50_ = 7.1 µM. The inhibition of linoleic acid oxidation by β-carotene bleaching systems, IC_50_ = 5.8 µM [175].	Dose-dependent inhibition of *S. aureus* and *C. albicans* biofilm formation [107].
	At concentrations up to 200 µg/mL, nontoxic in *C. elegans* model, slightly reduced nematode survival at 500 µg/mL [107].			
**Propolin F** **= isonymphaeol B**	12 µM inhibited cell proliferation in RAW 264.7 cells [174].	Inhibition of albumin denaturation, IC_50_ = 0.4 µM. Inhibition of nitrite production stimulated by LPS in RAW 264.7 cells, IC_50_ = 6.2 µM. COX-2 inhibitor = 23.8 µM [174].	DPPH radical scavenging activity, IC_50_ = 8.5 µM. The inhibition of linoleic acid oxidation by β-carotene bleaching systems, IC_50_ = 5.9 µM [175].	Dose-dependent inhibition of *S. aureus* biofilm formation. Inhibition of *C. albicans* biofilm formation [107].
**Propolin G** **= nymphaeol C**	At tested concentrations did not inhibit cell proliferation; it induced cell proliferation to some extent in RAW 264.7 cells [174].	Inhibition of albumin denaturation, IC_50_ = 0.37 µM. Inhibition of nitrite production stimulated by LPS in RAW 264.7 cells, IC_50_ = 2.4 µM. COX-2 inhibitor = 15.5 µM [174].	DPPH radical scavenging activity IC_50_ = 9.8 µM. The inhibition of linoleic acid oxidation by β-carotene bleaching systems, IC_50_ = 10.3 µM [175].	Inhibition of *C. albicans* biofilm formation [107].
**Scandenone** **= warangalone**	At 10 µM (86.5 % cell viability) in RAW 264.7 cells [119].	Inhibition of LPS-stimulated NO production in RAW 264.7 cells, IC_50_ = 8.5 µM [119].		
	20 µM inhibited proliferation in MDA-MB-231 cells and 15 µM in MCF-7 cells. Promotion of proliferation in MCF-10A [235].	Anti-inflammatory and antinociceptive activity in carrageenan-induced hind paw oedema model and TPA-induced mouse ear oedema model at 100 mg/kg dose [236].		
**Sophoraflavanone G**	Toxic from 4 to 64 µg/mL in human PBMC with >50% cellular activity inhibition. IC_50_ = 3.2 µg/mL [123].	COX-1 inhibitor IC_50_ = 0.1–0.6 µM. 5-LOX inhibitor IC_50_ = 0.1–0.3 µM. 12-LOX inhibitor IC_50_ = 20 µM [191].		Additive effect with ciprofloxacin, erythromycin, gentamicin, fusidic acid, oxacillin [123].
		Interruption of the NF-κB and MAPK signalling pathways [237].		
	Inhibition of cell proliferation in: A549, NCI-H460, SK-OV-3, SK-MEL-2, XF498, HCT-15, HL60, SPC-A-1 cells with IC_50_ = 2–36 μg/mL [238].	Inhibition of PGE_2_ production in LPS-induced RAW cells by COX-2 downregulation at 1–50 μM. 10–250 µg/ear in mouse croton oil-induced ear oedema and 2–250 mg/kg in rat carrageenan paw oedema → effect far less than prednisolone, but higher when applied topically [69].		Synergy with ampicillin and oxacillin [125].
	IC_50_ = 12.5 μM in HL-60 cells [239].	Inhibition of NO, PGE_2_, IL-1β, IL-6, TNFα production in Ag I/II-N-stimulated RAW 264.7 cells via the downregulation of iNOS and COX-2 expression. Inhibition of the phosphorylation of IκB-α, nuclear translocation of p65, and subsequent activation of NF- κB. Inhibition of MAPK-mediated pathways [240].		
	IC_50_ = 12.5 μM in HL-60 cells. IC_50_ = 13.3 μM in HepG2 cells [241].			
	CC_50_ =< 19 μM in HSC-2 cells. CC_50_ = 19 μM in HSG cells. CC_50_ = 19 μM in HGF cells [242].			
**Tomentodiplacone B**	IC_50_ = >20 μM in THP-1 cells, viability at 30 µM [232].	Reduction of TNF-α secretion as much as or more than the prednisone. IC_50_ >20 μM [232].	TEAC_ABTS_ = 1.0, TEAC_Inhibition of. peroxynitrite-induced tyrosine nitration_ *=* 0.8 [171].	
**Xanthoangelol**	IC_50_ = 23.6 μM in THP-1. IC_50_ = 21.7 μM in MRC-5. IC_50_ = 21.5 μM in HEK293. IC_50_ = 13.7 μM in HepG2. IC_50_ = 58.8 μM in CLS-54. IC_50_ = 1.0 μM in MRSA [109].	Inhibition of LPS-stimulated NO production, IC_50_ = 5 μM. Suppression of AP-1. Reduction of the phosphorylation (at serine 536) level of the p65 subunit of NF-κB [243].		
	IC_50_ = 25 μM in MRC-5. IC_50_ = 25 μM in THP-1 [108].			
**Xanthohumol**	IC_50_ = 5.2 μg/mL in NHLF cells [148].	NF-κB activity was reduced in vitro as well as in hepatic tissue after ischemia/reperfusion [244].	TEAC_ABTS_ = 0.3 μmol/l TEAC_FRAP_ = 0.3 μmol/l [245].	Synergy with oxacillin against *S. aureus* [147].
	IC_50_ = 11.0 μM in MCF-7. IC_50_ = 10.7 μM in PC-3. IC_50_ = 91.3 μM in HT-29 [246].	10 μg/mL inhibits 91.7% of the NO production by suppressing iNOS induced by a combination of LPS and IFN-γ [247].	DPPH radical scavenging activity, IC_50_ = 2.0 µM [246].	At MIC reduced biofilm viability by 86.5% [147].
	IC_50_ = 40.8 μM in HCT116. IC_50_ = 50.2 μM in HT-29. IC_50_ = 25.4 μM in HepG2. IC_50_ = 37.2 μM in Huh7 [248].	↓ the expression of the LPS receptor components - TLR4 and MD2 → suppression of NF-κB activation in LPS-activated RAW 264.7 cells. Inhibition of the binding activity of STAT-1alpha and IRF-1 In IFN-γ-stimulated RAW 264.7 cells [249].	TEAC_ABTS_ = 0.2, IC_50_ = 0.7 mg/mL, TEAC_DPPH_ = 0.04, IC_50_ = >1.2 mg/mL [250].	15–30 μg/mL → ↓ release of planktonic bacteria from the 24 h old biofilm by more than 90%. BBC = 60–125 μg/mL [148].
	IC_50_ = 3.6 μM in HCT-15 [251].	↓ the release of MCP-1 and TNF-α in LPS-stimulated RAW 264.7 and U937 human monocytes [252].	↓ reactive oxygen species in vitro. Levels of enzymatic and nonenzymatic antioxidants were restored after pretreatment in postischemic hepatic tissue, and lipid peroxidation was attenuated [244].	
	0.5–10 µM inhibited melanogenesis induced by isobutyl-methylxanthine in B16 melanoma cells [253].	Inhibition of IL-12 production in stimulated macrophages through the downregulation of NF-κB. In an oxazolone-induced chronic dermatitis model in mouse ear → attenuated dermatitis [254].		
		Stout beer supplemented with 10 mg/L of xanthohumol for 4 weeks decreased serum VEGF levels (18.4%), N-acetylglucosaminidase activity (27.8%), IL1β concentration (9.1%), and NO released (77.1%), accompanied by a reduced redox state as observed by an increased GSH/GSSG ratio (to 198.8%) [255].		
**3′-*O*-methyldiplacol**	IC_50_ = 7.2 μM in THP-1 cells [232].	Inhibition of LPS-induced NO production in RAW 264.7 cells, IC_50_ = 5.9 µM [173].	TEAC_ABTS_ = 1.6, TEAC_DPPH_ = 0.1, TEAC_FRAP_ = 0.1, TEAC_Inhibition of. peroxynitrite-induced tyrosine nitration_ *=* 0.7 [171].	Synergy with oxacillin. Additive effect with ciprofloxacin, tetracycline against MRSA [231].
**3′-*O*-methyldiplacone**	IC_50_ = 30.2 µM in WB 344 [164].	↓ the secretion of TNF-α ≥ than the prednisone [164].	DPPH quenching activity TEAC 0.8 at 10 µM [164].	
	IC_50_ =< 10 μM in THP-1 cells [232].		TEAC_ABTS_ = 1.4, TEAC_DPPH_ = 0.1, TEAC_FRAP_ = 0.1, TEAC_Inhibition of. peroxynitrite induced tyrosine nitration_ *=* 0.8 [171].	
	EC_50_ =< 10 µM in MCF-7 cells, EC_50_ =< 10 µM in CEM cells, EC_50_ = 7.3 µM in RPMI8226 cells, EC_50_ = 5.5 µM in U266 cells, EC_50_ = 7.4 µM in HeLa cells, EC_50_ = 4.7 µM in BJ cells, EC_50_<10 µM in THP-1 cells [172].			
**3′-*O*-methyl-5′-hydroxydiplacone**	Antiproliferative (IC_50_ = 12.6 μM) and cytotoxic (LC_50_>30 µM) effect [169].	Inhibition of LPS-induced NO production in RAW 264.7 cells, IC_50_ = 1.5 µM [173].	TEAC_ABTS_ = 1.7, TEAC_DPPH_ = 1.0, TEAC_FRAP_ = 0.7, TEAC_Inhibition of. peroxynitrite induced tyrosine nitration_ *=* 0.8, Superoxide scavenging activity-Enzymatic = 71.2%, Non-Enzymatic = 29.5% at 50 μM [171].	
		COX-1 inhibitor IC_50_ = 3.3 μM COX-2 inhibitor IC_50_ = 10.6 μM 5-LOX inhibitor IC_50_ = 0.1 μM [168].		
**3′-*O*-methyl-5′-*O*-methyldiplacone**	IC_50_ = 7.9 μM in THP-1 cells [232].	5-LOX inhibitor IC_50_ = 0.4 μM [168].	TEAC_ABTS_ 1.6, TEAC _DPPH_ = 0.3, TEAC _FRAP_ = 1.2, TEAC _Inhibition of. peroxynitrite-induced tyrosine nitration_ *=* 0.8 [171].	
**4-hydroxylonchocarpin**	IC_50_ > 100 μM in RAW 264.7 cells [256].	Inhibition (66.5%) of NO release from LPS-stimulated RAW 264.7 cells. 10 μM inhibited iNOS activity. In the carrageenan-induced paw oedema model, 10 mg/kg showed comparable activity to indomethacin, and 50 mg/kg showed higher activity than indomethacin [256].		

**Abbreviations:** reduction/decrease (↓), increase (↑) 5-LOX (5-lipoxygenase), A20 (ubiquitin-editing molecule), A23187 (calcium ionophore), *AHR* (aryl hydrocarbon receptor), AIM2 (interferon-inducible protein), Akt (protein kinase B), AP-1 (activator protein 1), ARE (antioxidant response element), ASC (caspase recruitment domain), BBC (biofilm bactericidal concentration), BUN (blood urea nitrogen), CAT (catalase), CC_50_ (50% cytotoxic concentration), CCl_4_ (tetrachlormethan), CCL5 (chemokine (C-–C motif) ligand 5), CCL17 (CC chemokine ligand 17), CCL22 (CC chemokine ligand 22), CD3 (cluster of differentiation 3), CD4+T-cell (T helper cells), c-Jun (Jun proto-oncogene), COX-1 (cyclooxygenase-1), COX-2 (cyclooxygenase-2), cPLA2 (cytosolic phospholipase A2), DCFH (dichloro-dihydro-fluorescein), DPPH (2,2-difenyl-1-pikrylhydrazyl), *E. coli* (*Escherichia coli*), EC_50_ (half-maximal effective concentration), ED_50_ (median effective dose), eNOS (endothelial nitric oxide synthase), ERK1/2 (extracellular signal-regulated kinases), ERK1/2/p90 (extracellular signal-regulated kinases), Fyn (proto-oncogene tyrosine-protein kinase), *Gata-3* (gene-GATA binding protein 3), GPx (glutathione peroxidase), GPx1 (glutathione peroxidase 1), HO-1 (heme oxygenase-1), *IBS* (*irritable bowel* syndrome), ICAM-1 (intercellular adhesion molecule-1), IFN-γ (interferon gamma), IL-10 (interleukin 10), IL-12 (interleukin 12), IL-12p40 (interleukin 12 subunit p40), IL-17 (interleukin 17), IL-1β (interleukin 1β), IL-2 (interleukin 2), IL-23 (interleukin 23), IL-4 (interleukin 4), IL-6 (interleukin 6), iNOS (inducible NO synthase), IRF-1 (interferon regulatory factor 1), IκB (inhibitor of κB), IκBα (nuclear factor of kappa light polypeptide gene enhancer in B-cell inhibitor alpha), JNK (Jun N-terminal kinase), KCa3 (1 calcium-activated potassium channel), Kv1.3 (voltage-gated potassium channel), LC_50_ (lethal concentration 50), LPS (lipopolysaccharide), LTC4 (leukotriene C4), MAPK (mitogen-activated protein kinase), MBIC (minimum inhibitory biofilm concentration), MCP-1 (monocyte chemoattractant protein-1), MD-2 (myeloid differentiation factor 2), MDA (malondialdehyde), MDC (macrophage-derived chemokine), MEK (mitogen-activated protein kinase), MMP-1 (matrix metalloproteinase-1), MMP-13 (matrix metalloproteinase-13), MMP-2 (matrix metalloproteinase-2), MMP-3 (matrix metalloproteinase-3), mPGES-1 (microsomal prostaglandin E synthase-1), mRNA (messenger RNA), MRSA (methicillin-resistant *Staphylococcus aureus*), MSSA (methicillin-sensitive *Staphylococcus aureus*), MSU (monosodium urate crystals), mTOR (Akt/mammalian target of rapamycin), NADPH (nicotinamide adenine dinucleotide phosphate), NEK7 (NIMA-related kinase 7), NF-κB (nuclear factor-κB), NF-κB p65 (subunit of NF-kappa-B transcription complex), NLRC4 (NLR family CARD domain containing 4), NLRP3 (NLR family pyrin domain containing 3), NO (nitric oxide), NOS (NO synthase), NOX (NADPH oxidase activity), Nrf2 (nuclear factor erythroid 2-related factor 2), ORAC (oxygen radical absorbance capacity), ORAI1 (calcium release-activated calcium channel protein 1), ox-LDL (oxidized low-density lipoprotein), *P. acnes* (*Propionibacterium acnes*), *P. aeruginosa* (*Pseudomonas aeruginosa*), PBS (phosphate-buffered saline), PCA (passive cutaneous anaphylaxis), PGE2 (prostaglandin E_2_), PI3K (phosphoinositide 3-kinase), PKA (protein kinase A), PMA (phorbol-12-myristate-13-acetate), PPAR-α (peroxisome proliferator-activated receptor alpha), pro-MMP-2 (promatrix metalloproteinase-2), RANTES (regulated upon activation, normal T cell expressed and presumably secreted), ROS (reactive oxygen species), SAPK/JNK (stress-activated protein kinase/c-Jun-*N*-terminal kinase), SC_50_ (scavenging DPPH free radicals by 50%), SC_50_ (scavenging DPPH free radicals by 50%), SEA (*Staphylococcus aureus* enterotoxin), SEB (*Staphylococcus aureus* enterotoxin B), SOD (superoxide dismutase)**,** SOD1 (superoxide dismutase 1)**,** SOD2 (superoxide dismutase 2), STAT1 (signal transducer and activator of transcription 1), STAT3 (signal transducer and activator of transcription 3), STAT6 (signal transducer and activator of transcription 6), sUV (solar ultraviolet), TARC (thymus- and activation-regulated chemokine), t-BuOOH (tert-butyl hydroperoxide), TEAC (trolox equivalent antioxidant capacity), TEAC_ABTS_ (trolox equivalent antioxidant capacity 2,2’-azino-bis(3-ethylbenzothiazoline-6-sulfonic acid)), TLR (toll-like receptor), TLR4 (toll-like receptor 4), TNCB (2,4,6-trinitrochlorobenzene), TNF-α (tumour necrosis factor α), TPA (12-*O*-tetradecanoylphorbol acetate), VCAM-1 (vascular cell adhesion molecule 1), VEGF (vascular endothelial growth factor). **Cell lines:** 1A9 (endometrioid ovary carcinoma), A549 (adenocarcinomic human alveolar basal epithelial cells), *AML12* (murine hepatocyte), Bcap37 (breast cancer), BEAS-2B (bronchial epithelial cells), BJ (fibroblasts), C6 (glioma), CAKI-1 (renal cancer), CCRF-CEM (acute lymphoblastic leukaemia), CEM (acute lymphoblastic leukaemia), CLS-54 (lung adenocarcinoma), H4IIE (hepatoma), HaCaT (aneuploid immortal keratinocyte), HCT116 (colorectal carcinoma), HCT-15 (colorectal carcinoma), HCT-8 (ileocecal carcinoma), HDFs (human dermal fibroblasts), HEK293 (embryonic kidney cells), HeLa (cervical cancer), Hep3B (hepatocellular carcinoma), HepG2 (hepatocellular carcinoma), HGF (primary gingival fibroblast), HL60 (acute promyelocytic leukaemia), HPC (hematopoietic progenitor cell), HPLF (periodontal ligament fibroblasts), HSC-2 (oral squamous cell carcinoma), HSG (submandibular gland), HT-29 (colorectal adenocarcinoma), Huh7 (hepatocellular carcinoma), HUVEC (human umbilical vein endothelial cells), K562r (chronic myeloid leukaemia at blast crisis), K562s (chronic myeloid leukaemia at blast crisis), KB-VIN (epidermoid carcinoma of the nasopharynx and its subclone), L-02 (papillomavirus-related andocervical adenocarcinoma), L-929 (fibroblasts), LNCaP (prostate adenocarcinoma), MC/9 (foetal liver mast cell), MCF-10A (breast epithelial cells), MCF-7 (breast cancer), MDA-MB-231 (breast adenocarcinoma), MDA-MB-468 (breast cancer), MOLM-13 (acute myeloid leukaemia), MRC-5 (fibroblasts), NB4 (acute promyelocytic leukaemia), NCI-H460 (non-small-cell lung cancer), NHLF (lung fibroblasts), PBMC (primary peripheral blood mononuclear cells), PC3 (Caucasian prostate adenocarcinoma), RAW 264.7 (macrophages), RPMI8226 (multiple myeloma), SK-MEL-2 (melanoma), SK-OV-3 (cystadenocarcinoma), SPC-A-1 (lung adenocarcinoma), SW480 (colorectal adenocarcinoma), SW-620 (adenocarcinoma), SZ95 (sebocytes), Tca8113 (squamous cell carcinoma of the tongue), THP-1 (acute monocytic leukaemia), U251 (malignant glioblastoma), U266 (multiple myeloma), U937 (histiocytic lymphoma), Vero (kidney epithelial cells), WB-F344 (epithelial cells), XF498 (glioblastoma), ZR-75-1 (breast cancer).

## Data Availability

No new data were created or analysed in this study. Data sharing is not applicable to this article.

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
