# Peer review of "Prenylated Flavonoids in Topical Infections and Wound Healing"

_molecules, 2022, doi:10.3390/molecules27144491_

Round 1

Reviewer 1 Report

The work entitled: Prenylated Flavonoids in Topical Infections and Wound Healing is an extensive and adequate review of the subject.

Line 28: predominantly S. aureus strains,… Do not abbreviate the name of the microorganism the first time it is mentioned. Also, in line 106 it is used without abbreviation: are caused by Staphylococcus aureus,…

Line 32: in vitro and in vivo. Italic letters for both terms

CLSI or EUCAST, MIC and MBC. All abbreviations must be defined on first use.

Line 94: Define ROS

Line 125: Define MRSA

Line 127: As in vitro studies… in vitro in italic letter

Line 159: Define PUFA, define all abbreviations in the script.

Paragraphs, lines 183-247: This section has very long paragraphs. You need to separate by ideas. Please separate into clear paragraphs.

Line 235: of Hou et al. (2013), Change for of Hou et al. [47]. Correct throughout the manuscript.

In general, all the paragraphs are very long and that makes it difficult to read the article. It can be boring for the readers and they would lose interest in it. Paragraphs should be between 15-20 lines maximum, please correct this throughout the manuscript.

Lines 574-576…MICs (< 0.31‒9.76 µg/mL) [130]. A similar spectrum of microorganisms was used in the study by Mbaveng et al. (2008), where isobavachalcone strongly inhibited the growth of pathogens with MICs of 0.3‒1.2 µg/mL …. Please unify the way of expressing the figures. Let there be two significant figures in all the data represented. Correct throughout the manuscript.

Table 2 has too much information and should be presented in a more appropriate way.

Best regards,

Author Response

Dear reviewer, thank you for the time you paid to review our manuscript, as well as for all your suggestions on how to improve the text.

Line 28: predominantly S. aureus strains,… Do not abbreviate the name of the microorganism the first time it is mentioned. Also, in line 106 it is used without abbreviation: are caused by Staphylococcus aureus,…

The name of bacteria was extended in the abstract and abbreviated in the text.

Line 32: in vitro and in vivo. Italic letters for both terms

Done.

CLSI or EUCAST, MIC and MBC. All abbreviations must be defined on first use.

Done. Please, see the abstract.

Line 94: Define ROS

Done.

Line 125: Define MRSA

Done (Line 109).

Line 127: As in vitro studies… in vitro in italic letter

Corrected throughout the whole text.

Line 159: Define PUFA, define all abbreviations in the script.

All abbreviations were corrected throughout the whole text.

Paragraphs, lines 183-247: This section has very long paragraphs. You need to separate by ideas. Please separate into clear paragraphs.

Done.

Line 235: of Hou et al. (2013), Change for of Hou et al. [47]. Correct throughout the manuscript.

Corrected throughout the whole text.

In general, all the paragraphs are very long and that makes it difficult to read the article. It can be boring for the readers and they would lose interest in it. Paragraphs should be between 15-20 lines maximum, please correct this throughout the manuscript.

Subsections have been created throughout the text where it was relevant. The discussion remained unchanged due to its complexity.

Lines 574-576…MICs (< 0.31‒9.76 µg/mL) [130]. A similar spectrum of microorganisms was used in the study by Mbaveng et al. (2008), where isobavachalcone strongly inhibited the growth of pathogens with MICs of 0.3‒1.2 µg/mL …. Please unify the way of expressing the figures. Let there be two significant figures in all the data represented. Correct throughout the manuscript.

We are very sorry, but we are not sure if we clearly understand the request. If you meant numbers – we have uniformed them, when it was relevant through the whole manuscript.

Table 2 has too much information and should be presented in a more appropriate way.

The aim of the second part of the review was to add multiple beneficial activities of prenylated flavonoids involved in wound healing. Tracking down these activities was very challenging, and processing such a large amount of data into a table seemed to us to be the easiest way to present the most important information. We consider these data relevant for future research on prenylated flavonoids and the table in this form highlights only promising candidates. However, due to the scope of the text, only a few structures, that met the selection criteria defined in the text were discussed in more detail. According to our opinion, all information in table 2 is very important for the complex overview. We are afraid, that if we delete something, it might be missing in context. Furthermore, if we flip over the table to text, it may become boring for readers.

Thank you so much.

Reviewer 2 Report

This is a very interesting and comprehensive review on the supporting effect of flavonoid compounds on wound healing especially in the context of their antimicrobial activity. The topic is worthy of the investigation because the increased interest in natural compounds is observed. The significance of the work has been properly highlighted and the relevant background has been provided. Data is compiled in the form of extensive tables. I have only a few minor suggestions of editorial nature:

1)      All abbreviations should be explained when they are used for the first time in the text (see: e.g MRSA – in line 107; EOs – line 158, EGCG – line 208, MSSA – line 440)

2)      „in vivo”, „in vitro” should be italized

3)      Fonts on Figures 1 and 2 should be larger to better visibility

4)      What does it mean red part of structures presented on Figs 3-12 ? The explanation should be given in Figure legends.

Good work. Congratulation!

Author Response

Dear reviewer, thank you very much for the time you paid to review our manuscript, as well as for all your suggestions on how to improve the text.

  • All abbreviations should be explained when they are used for the first time in the text (see: e.g MRSA – in line 107; EOs – line 158, EGCG – line 208, MSSA – line 440).

Corrected throughout the whole text.

  • „in vivo”, „in vitro” should be italized

Corrected throughout the whole text.

  • Fonts on Figures 1 and 2 should be larger to better visibility

The Figures were corrected to be more visible.

  • What does it mean red part of structures presented on Figs 3-12? The explanation should be given in Figure legends.

A better explanation was added to the text. Please see lines 485-486: “The red highlighted parts of molecules indicate the difference between similar structures and might explain the structure-antibacterial activity relationship.”

Good work. Congratulation!

Thank you so much!

Round 2

Reviewer 1 Report

Thanks to the authors for responding to the comments. I have no more comments. I think it can be accepted in the current format.

Best regards,